# TEMPORAL-DIFFERENCE LEARNING FOR NONLINEAR VALUE FUNCTION APPROXIMATION IN THE LAZY TRAINING REGIME

## ABSTRACT

We discuss the approximation of the value function for infinite-horizon discounted Markov Reward Processes (MRP) with nonlinear functions trained with the Temporal-Difference (TD) learning algorithm. We consider this problem under a certain scaling of the approximating function, leading to a regime called lazy training. In this regime the parameters of the model vary only slightly during the learning process, a feature that has recently been observed in the training of neural networks, where the scaling we study arises naturally, implicit in the initialization of their parameters. Both in the under- and over-parametrized frameworks, we prove exponential convergence to local, respectively global minimizers of the above algorithm in the lazy training regime. We then give examples of such convergence results in the case of models that diverge if trained with non-lazy TD learning, and in the case of neural networks.

## 1 INTRODUCTION

In recent years, deep reinforcement learning has pushed the boundaries of Artificial Intelligence to an unprecedented level, achieving what was expected to be possible only in a decade and outperforming human intelligence in a number of highly complex tasks. Paramount examples of this potential have appeared over the past few years, with such algorithms mastering games and tasks of increasing complexity, from playing Atari to learning to walk and beating world grandmasters at the game of Go (Haarnoja et al., 2018; Mnih et al.; 2013; Silver et al., 2016; 2017; 2018). Such impressive success would be impossible without using neural networks to approximate value functions and / or policy functions in reinforcement learning algorithms. While neural networks, in particular deep neural networks, provide a powerful and versatile tool to approximate high dimensional functions (Barron, 1993; Cybenko, 1989; Hornik, 1991), their intrinsic nonlinearity might also lead to trouble in training, in particular in the context of reinforcement learning. For example, it is well known that nonlinear approximation of the value function might cause divergence in classical temporal-difference learning due to instability (Tsitsiklis & Van Roy, 1997). Several algorithms have been proposed in the literature to address the issue of non-convergence (Bhatnagar et al., 2009; Maei & Sutton, 2010; Riedmiller, 2005; Sutton et al., 2009a;b; Szepesvári, 2010), while practical deep reinforcement learning often employs and prefers basic algorithms such as temporal-difference (Sutton, 1988) and Q-learning (Watkins, 1989) due to their simplicity. It is thus crucial to understand the convergence of such algorithms and to bridge the gap between theory and practice.

The theoretical understanding of deep reinforcement learning is of course rather challenging, as even for supervised learning, which can be viewed as a special case of reinforcement learning, deep neural networks are still far from being understood despite the huge amount of research focus in recent years. On the other hand, recent progress has led to an emerging theory for neural network learning at least in the regime of over-parametrization, including recent works on mean-field point of view of training dynamics (Chizat & Bach, 2018a; Mei et al., 2018; Rotskoff et al., 2019; Rotskoff & Vanden-Eijnden, 2018; Wei et al., 2018) and also the linearized training dynamics in the over-parametrized regime (Allen-Zhu et al., 2018a;b; Chizat & Bach, 2018b; Du et al., 2018a;b; Ghorbani et al., 2019a; Jacot et al., 2018; Lee et al., 2019; Oymak & Soltanolkotabi, 2019; Zou et al., 2018).

The main goal of this work is to analyze the dynamics of a prototypical reinforcement learning algorithm – temporal/difference (TD) learning – based on the recent progress in deep supervised learning. In particular, we will focus on the lazy training regime, inspired by the recent work Chizat

& Bach (2018b), and analyze TD learning in both over-parametrized and under-parametrized regimes with scaled value function approximations.

**Related Works.** This work is closely related to the recent paper Chizat & Bach (2018b), addressing the problem of lazy training in the supervised learning framework when models are trained through (stochastic) gradient descent. In particular, that paper introduced the scaling that we consider in this work as an explanation, *e.g.*, of the small relative displacement of the weights of over- and under-parametrized neural networks for supervised learning. That work, however, leverages the gradient structure of the underlying vector field, which we lack in the present framework when the underlying policy is not reversible (Ollivier, 2018). The linear stability analysis is also considered in the recent work Achiam et al. (2019) based on the neural tangent kernel (Jacot et al., 2018) for off-policy deep Q-learning.

The groundbreaking paper Tsitsiklis & Van Roy (1997) proves convergence of TD learning for linear value function approximation, unifying the manifold interpretations of this convergence phenomenon that preceded it by highlighting that convergence of the algorithm is to be understood in the norm induced by the invariant measure of the underlying Markov process. Furthermore, the paper gives an illuminating counterexample for the extension of the linear result to the general, nonlinear setting. Our result shows that divergence does not occur in the lazy training regime.

Concurrent work (Brandfonbrener & Bruna, 2019) has shown convergence and non-divergence of TD learning in the over-parameterized, respectively the under-parametrized regime, provided that the environment is sufficiently reversible. We note that working in the lazy training regime allows to ensure convergence independently on the reversibility of the environment and quantify the error of the fitted model in the under-parametrized regime. Finally, another concurrent work (Cai et al., 2019) analyzes global convergence of a modified TD algorithm for two-layer neural networks with ReLu nonlinearity when the width of the hidden layer diverges. In contrast, in the present paper we focus on the original TD($\lambda$) learning algorithm for general approximators.

**Contributions.** This paper proves that on-policy TD learning for policy evaluation (on-policy policy-evaluation for short), a widely used algorithm for value function approximation in reinforcement learning, is convergent (asymptotically with probability one), in the lazy training regime, when the model is a *nonlinear* function of its parameters. More specifically, we prove convergence of this algorithm in both the under- and over-parametrized regime to local and global minima, respectively, of a natural, weighted error function (the projected TD error), and illustrate such convergence properties through numerical examples.

To obtain the result summarized above, we adapt the contraction conditions developed in the framework of linear function approximations to a nonlinear, differential geometric setting. Furthermore, we extend some existing results on the convergence in the lazy training regime of nonlinear models trained by gradient descent in the supervised learning framework to the world of reinforcement learning. This requires a generalization of the techniques developed in the gradient flow setting to non-gradient (*i.e.*, rotational) vector fields such as the ones encountered in the TD learning framework.

## 2 MARKOV DECISION PROCESSES

We denote a Markov Reward Process (MRP) by the 4-tuple $(\mathcal{S}, P, r, \gamma)$, where $\mathcal{S}$ is the state space, $P = P(s, s')_{s,s' \in \mathcal{S}}$ a transition kernel, $r(s, s')_{s,s' \in \mathcal{S}}$ is the real-valued, bounded immediate reward function and $\gamma \in (0, 1)$ is a discount factor. In this context, the *value function $V : \mathcal{S} \to \mathbb{R}_+$* maps each state to the infinite-horizon, expected discounted reward obtained by following the Markov process defined by $P$. We assume that this Markov process satisfies the following assumption:

**Assumption 1.** *The Markov process with transition kernel $P$ is ergodic and its stationary measure $\mu$ has full support in $\mathcal{S}$. Furthermore we assume that $\mathcal{S}$ is compact.*

In this note we are interested in learning the value (or cost-to-go) function $V^*(x)$ of a given MRP $(\mathcal{S}, P, r, \gamma)$, which is given by

$$V^*(s) := \mathbb{E}_s \left[ \sum_{t=0}^{\infty} \gamma^t r(s_t, s_{t+1}) \right], \tag{1}$$

where $\mathbb{E}_s [\cdot]$ denotes the expectation of the stochastic process $s_t$ starting at $s_0 = s$. More specifically we would like to estimate this function through a set of predictors $V_w(s)$ in a Hilbert space $\mathcal{F}$ parametrized by a vector $w \in \mathcal{W} := \mathbb{R}^p$. We make the following assumption on such predictors:

**Assumption 2.** *The parametric model $V : \mathbb{R}^p \to \mathcal{F}$ mapping $w \mapsto V_w(\cdot)$ is differentiable with Lipschitz continuous derivative $DV : w \mapsto DV_w$ (where $DV_w$ is a linear map from $\mathbb{R}^p \to \mathcal{F}$) with Lipschitz constant $L_{DV}$ defined* WRT *the operator norm.*

A popular algorithm to solve this problem is given by value function approximation with *TD($\lambda$)* updates (Sutton & Barto, 2018). Starting from an initial condition $w(0) \in \mathcal{W}$, for any $\lambda \in [0, 1)$, this learning algorithm updates the parameters $w$ of the predictor by the following rule:

$$w(t + 1) := w(t) + \beta_t \delta(t) z_\lambda(t) , \tag{2}$$

for a *fixed* sequence of time steps $\{\beta_t\}$ to be specified later, where the *temporal-difference error* $\delta(t)$ and *eligibility vector* $z_\lambda(t)$ are given by

$$\delta(t) := r(s_t, s_{t+1}) + \gamma V_{w(t)}(s_{t+1}) - V_{w(t)}(s_t) \qquad z_\lambda(t) := \sum_{\tau=0}^{t} (\gamma\lambda)^{t-\tau} \nabla_w V_{w(t)}(s_\tau) . \tag{3}$$

This work focuses on the asymptotic regime of small constant step-sizes $\beta_t \to 0$. In this adiabatic limit, the stochastic component of the dynamics is averaged out before the parameters of the model can undergo a significant change. This allows to consider the TD update as a deterministic dynamical system emerging from the averaging of the underlying stochastic algorithm. We focus on analysis of this deterministic system to highlight the aspect of nonlinear function approximation. The averaged, deterministic dynamics is given by the set of ODEs

$$\frac{\mathrm{d}}{\mathrm{d}t} w(t) = \mathbb{E}_\mu \left[ \left( r(s, s') + \gamma V_{w(t)}(s') - V_{w(t)}(s) \right) z_\lambda(t) \right] , \tag{4}$$

where $\mathbb{E}_\mu$ denotes the expectation with respect to the invariant measure of the underlying dynamics. In the case of finite state space ($|\mathcal{S}| = d$) we can represent $V_w$ as a vector in $\mathbb{R}^d$, while in general it is a function $\mathcal{S} \to \mathbb{R}$, which we will restrict to the space $L^2(\mathcal{S}, \mu)$, namely square integrable function with respect to the measure $\mu$.

To streamline our analysis of the TD algorithm, we define the TD operator $T^\lambda : L^2(\mathcal{S}, \mu) \to L^2(\mathcal{S}, \mu)$:

$$T^\lambda V(s) := (1 - \lambda) \sum_{m=0}^{\infty} \lambda^m \mathbb{E}_s \left[ \sum_{t=0}^{m} \gamma^t r(s_t, s_{t+1}) + \gamma^{m+1} V(s_{m+1}) \right] .$$

Note that when $\lambda = 0$ the above operator acquires the simple form $T^0 V := \bar{r} + \gamma P V$ for $\bar{r}(s) := \mathbb{E}_s [r(s, s')]$. Then, denoting throughout by $DV_w$ the Fréchet derivative of $V$ at $w$, it can be shown (Tsitsiklis & Van Roy, 1997, Lemma 8) (and is immediately verified in the special case $\lambda = 0$) that the continuous dynamics (4) for general $\lambda < 1$ can be written as

$$\frac{\mathrm{d}}{\mathrm{d}t} w(t) = \langle T^\lambda V_{w(t)} - V_{w(t)}, DV_{w(t)} \rangle_\mu , \tag{5}$$

where we define throughout the inner product induced by the invariant measure $\mu$ (acting component-wise in expressions such as the one above) as

$$\langle a, b \rangle_\mu := \int_\mathcal{S} a(s) b(s) \mu(\mathrm{d}s) , \tag{6}$$

and denote by $\| \cdot \|_\mu$ the corresponding norm. Note that in the case $|\mathcal{S}| = d$, denoting by $\Gamma$ the $d$-dimensional diagonal matrix whose entries are the (positive) values of the invariant measure $\mu(s)$, one has $\langle a, b \rangle_\mu = a^\top \Gamma b$. The extension of convergence results for the limiting, average dynamics we consider in this paper to convergence with probability one of the underlying, stochastic algorithm can be obtained through standard stochastic approximation arguments (Borkar & Meyn, 2000; Borkar, 2009). More details on this straightforward extension are given in Remark 3.4 in Section 3 and in the appendix.

In this work, we are interested in a certain scaling of the TD learning algorithm with function approximation. More specifically, we consider the rescaled update

$$\frac{\mathrm{d}}{\mathrm{d}t} w(t) = \frac{1}{\alpha} \langle T^\lambda(\alpha V_{w(t)}) - \alpha V_{w(t)}, DV_{w(t)} \rangle_\mu \tag{7}$$

for large values of the scaling parameter $\alpha > 1$. One of the reasons why this scaling of the model is of practical interest is because it arises naturally when training neural networks, implicit in some widely applied choices of initial conditions, as we explain in Section 4.2. Furthermore, as we shall see below, under some mild assumptions for large values of $\alpha$ the parameters $w$ of the model vary only slightly during training, inducing what is called the "lazy training" regime. A visual representation of the geometric effect of this scaling in the case where $p < d < \infty$ is given in Fig. 1.

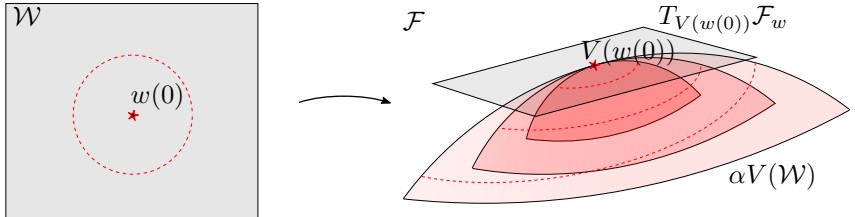

Figure 1: Schematic representation of the effect of the linear scaling of the approximating function (*e.g.*, in (11)) in the *under-parametrized* setting. The space of parameters (left) is mapped to the space of predictors (right) by the parametric model $V$. The scaling $V \to \alpha V$ changes the manifold $\mathcal{F}_w$ that the parameter space is mapped to (different surfaces on the right). In particular, this scaling "widens" the reach in the space of functions of the predictors within a ball of small radius in $\mathcal{W}$, but at the same time it "flattens" that space (locally in $\mathcal{W}$) bringing it closer to the tangential plane to the initial model $V_{w(0)}$. Choosing $V_{w(0)} = 0$ as in the picture above leaves the initial point of the dynamics (in predictor space) invariant under such transformation.

## 3 MAIN RESULTS

### 3.1 OVER-PARAMETRIZED REGIME

In the over-parametrized setting we assume that $DV_{w(0)}$ is surjective, *i.e.*, its singular values are uniformly bounded away from $0$. This is only possible in the finite state space setting and is automatically the case if the number of parameters $p$ is larger than the size of the state space $\mathcal{S}$. Admittedly, in applications such as AlphaGo (Silver et al., 2016; 2017), it is unrealistic to over-parametrize, but we start with this regime as it parallels the study of over-parametrized supervised learning for global convergence of the training loss. Analysis of the under-parametrized regime will be discussed in the next subsection. In order to state our first result, we introduce the scalar product in $\mathcal{F}$ defined by $\langle a, b \rangle_0 = \langle a, g_{w(0)} b \rangle$ where $g_w := (DV_w \cdot DV_w^\top)^{-1}$, and denote by $\| \cdot \|_0$ the norm it induces. Note that $g_w$ is the metric tensor associated to the pushforward metric induced by the parametric model $V : \mathbb{R}^p \to \mathcal{F}$. We note that if $DV_{w(0)}$ has singular values that are uniformly bounded away from $0$, the norms $\| \cdot \|_\mu, \| \cdot \|_0$ are equivalent, *i.e.*, there exists $\kappa > 0$ such that $\kappa^{-1} \|f\|_0 < \|f\|_\mu < \kappa \|f\|_0$ for all $f \in \mathcal{F}$.

**Theorem 3.1** (Over-parametrized case). *Assume that $\sigma_{min} > 0$, where $\sigma_{min}$ is the smallest singular value of $DV_{w(0)}$. Assume further that $w(0)$ is such that $\|V_{w(0)}\|_0 < M := (1 - \gamma)^2 \sigma_{min}^2 / (192 \kappa^2 L_{DV} \|DV_{w(0)}\|)$, then for $\alpha > \alpha_0 := \|V^*\|_0 / M$ we have for all $t \geq 0$ that*

$$\|V^* - \alpha V_{w(t)}\|_0^2 \leq \|V^* - \alpha V_{w(0)}\|_0^2 e^{-\frac{1-\gamma}{2\kappa^2} t} . \tag{8}$$

*Recall that $V^*$ is the exact value function given by (1). Moreover, if $\|V_{w(0)}\|_0 \leq C\alpha^{-1}$ for a constant $C > 0$, then $\sup_{t>0} \|w(t) - w(0)\| = \mathcal{O}(\alpha^{-1})$.*

Similarly to the proof in Chizat & Bach (2018b), we first show that $DV_w$ and $V_w$ do not change much assuming that $w$ stays in a small ball of radius $\varrho$. Then, combining this result with the Lipschitz continuous character of $DV$ in $w$, one shows that $w$ does indeed stay in the desired ball of radius $\varrho$. A similar computation can be done in our case. To bypass the absence of a strongly convex cost functional in our framework, which was crucial in the analysis of Chizat & Bach (2018b), we adopt a strategy based on the use of a local Lyapunov function

$$U(f) = \|f - V^*\|_0^2 , \tag{9}$$

where $V^*$ is the sought for value function (1). The theorem is based on some preparatory lemmas, proofs of which can be found in appendix. The first one states that for large values of the scaling parameter $\alpha$ the pushforward metric $g_w$ varies in a negligible way during training. Throughout, we denote by $\mathbb{1}$ the identity map in the corresponding space and by $\mathcal{B}_\varrho^\mu(v)$, $\mathcal{B}_\varrho^0(v)$ and $\mathcal{B}_\varrho(v)$ the balls with radius $\varrho$ around $v$ in $\| \cdot \|_\mu, \| \cdot \|_0$ and $\| \cdot \|_2$ respectively.

**Lemma 3.2** (Perturbation of the metric). *Let $\mathcal{G}_0$ be a compact subset of a linear space $\mathcal{G}$. For $v(0) \in \mathcal{G}_0$, let $g_v$ be a continuous, self-adjoint linear operator that is positive definite in a neighborhood of $v(0)$ when restricted on $\mathcal{G}$. Then for all $\varepsilon > 0$ there exists $\delta > 0$ such that, for all $v \in \mathcal{B}_\delta(v(0)) \subseteq \mathcal{G}_0$*

$$g_{v(0)} = (\mathbb{1} + \tilde{g}_v) g_v , \tag{10}$$

*for a linear operator $\tilde{g}_v$ with $\|\tilde{g}_v\| < \varepsilon$. More specifically, let $\sigma_{min}$ be the smallest singular value of $DV_{w(0)}$. Then if $\varrho \le (1-\gamma)\sigma_{min}^2/(48L_{DV})$, (10) holds with $\|\tilde{g}_{V(w)}\| < \frac{1-\gamma}{4}$ for all $w \in \mathcal{B}_\varrho(w(0))$.*

We also recall from Tsitsiklis & Van Roy (1997) the following contraction property of the TD operator in the $\| \cdot \|_\mu$ norm. For the convenience of readers, we recall the proof in the appendix.

**Lemma 3.3.** *(Tsitsiklis & Van Roy, 1997, Lemmas 1, 3, 7) Under Assumption 1, for any $V, \tilde{V} \in \mathcal{F}$ we have that $\|T^\lambda V - T^\lambda \tilde{V}\|_\mu \le \gamma_\lambda \|V - \tilde{V}\|_\mu$ for $\gamma_\lambda := \gamma\frac{1-\lambda}{1-\gamma\lambda} \le \gamma < 1$. In particular there exists a unique fixed point of $T^\lambda$, $V^* \in \mathcal{F}$ given by (1).*

The proof of Theorem 3.1 relies on the above lemma to establish decay of the local Lyapunov function $U$ as long as $w$ stays within a ball. The nonlinear effects become negligible when $\alpha$ is sufficiently large. The control of $U$ in turn gives the bound of the change of $w$, which closes the argument. The details are given in the supplementary materials.

**Remark 3.4.** *Our results can be extended to show stability and convergence in the stochastic approximation setting, similarly to Bhatnagar et al. (2009); Tsitsiklis & Van Roy (1997), under the additional assumption that the step size $\{\beta_t\}$ satisfies the Robbins-Monro condition (Robbins & Monro, 1951). For example, one can apply (Borkar & Meyn, 2000, Thms. 2.2, 2.4) guaranteeing almost sure convergence and exponential contraction of the expected error with probability one over the initial condition provided that the limiting vector field (in our case (7)) has a unique fixed point and is Lipschitz continuous. Lipschitz continuity is an immediate consequence of the linearity of $T^\lambda$ and the boundedness of closed balls in $\mathcal{F}$ together with the Lipschitz continuity of the models Assumption 2. The existence of a fixed point (1) in $\mathcal{F}$ of the limiting vector field is trivial while its uniqueness is shown in the proof of Theorem 3.1 in the appendix.*

### 3.2 UNDER-PARAMETRIZED REGIME

We now proceed to state and prove a convergence theorem in the under-parametrized case. The underlying assumption in this section is that the size of state space is larger than the number of parameters, which in turn bounds the rank $r$ of $DV_{w(0)}$ from above: $r < p < d$ (where possibly $d = \infty$). In this regime, in general, there is no hope that TD will converge to the true value function $V^*$. In fact, the image of the operator $T^\lambda$ might not even lie in the space $\mathcal{F}_w$ of approximating functions. However, the derivative $DV_{w(t)}^\top$ in the TD update acts as a projection (WRT the product $\langle \cdot, \cdot \rangle_\mu$) onto the tangent space of $\mathcal{F}_w$ at $V_{w(t)}$ (more specifically, $DV_{w(t)}^\top$ projects the image of $T^\lambda$ onto $\mathcal{W}$, which is then mapped back to $T_{V(w(t))}\mathcal{F}_w$ by $DV_{w(t)}$). We denote throughout by $\Pi$ and $\Pi_0$ the projection operator under (6) onto $T_{V(w(t))}\mathcal{F}_w$ and $T_{V(w(0))}\mathcal{F}_w$ respectively. What one can hope for is that the TD algorithm converges to a locally "optimal" approximation $\tilde{V}^*$ of $V^*$ on the manifold $\mathcal{F}_w$, which is close to the best approximator $\Pi_0 V^*$ of $V^*$ on the linear tangent space $T_{V(w(0))}\mathcal{F}_w$.

**Theorem 3.5** (Under-parametrized case)**.** *Assume that $r := rank(DV_w)$ is constant in a neighborhood of $w(0)$ and $V_{w(0)} = 0$. Then there exists $\alpha_0 > 0$ such that for any $\alpha > \alpha_0$ the dynamics (7) (and the corresponding approximation $V_w$) converge exponentially fast to a locally (in $\mathcal{W}$) attractive fixed point $\tilde{V}^*$, for which $\|\Pi(T^\lambda \tilde{V}^* - \tilde{V}^*)\|_\mu = 0$ and $\|\tilde{V}^* - V^*\|_\mu < \frac{1-\lambda\gamma}{1-\gamma}\|\Pi_0 V^* - V^*\|_\mu + \mathcal{O}(\alpha^{-1})$.*

Note that for random initialization the constant rank assumption is generically satisfied. Indeed, the maximal rank property holds generically in $\mathcal{W}$ and thus WP1 at $w(0)$ when the model parameters are initialized randomly. Furthermore, by the lower semicontinuity of the rank function the Jacobian $DV$ will have maximal rank in an *open* subset of $\mathcal{W}$. The main difference of the proof of the above result WRT the one in the over-parametrized regime is that $DV_w \cdot DV_w^\top$ does not have full rank anymore. This implies on one hand that the norms $\| \cdot \|_\mu$ and $\| \cdot \|_0$ are not equivalent in $\mathcal{F}$, even though we still have $\| \cdot \|_0 \le \kappa \| \cdot \|_\mu$ for a $\kappa > 0$, provided that Assumption 1 holds. On the other hand, as mentioned above, this implies that the model $V_w$ evolves on a submanifold $\mathcal{F}_w$ of $\mathcal{F}$, and that $T^\lambda$ does not, in general, map onto the tangential plane $T_{V(w)}\mathcal{F}_w$ of $\mathcal{F}_w$ at $V_w$. The action of $T^\lambda$ is then projected back onto $T_{V(w)}\mathcal{F}_w$ by the operator $DV_{w(t)}$. The nonlinear structure of the space $\mathcal{F}_w$ slightly complicates the proof WRT the over-parametrized case, and we apply standard differential geometric tools to map the problem back to a linear space.

$$
\begin{array}{ccc}
\mathcal{W}_0 & \xrightarrow{\ V\ } & \mathcal{F}_0 \\
\downarrow{\scriptstyle \phi} & & \downarrow{\scriptstyle \psi} \\
\overline{\mathcal{W}}_0 & \xrightarrow{\ \pi_r\ } & \overline{\mathcal{F}}_0
\end{array}
$$

*Proof.* We apply the rank theorem (Boutaib, 2015; Lee, 2003) ((Abraham et al., 2012) for the $\infty$-dimensional setting) to show that there exist sets $\mathcal{W}_0, \overline{\mathcal{W}}_0 \subseteq \mathbb{R}^p, \mathcal{F}_0, \overline{\mathcal{F}}_0 \subseteq \mathcal{F}$ and diffeomorphic maps $\phi : \mathcal{W}_0 \to \overline{\mathcal{W}}_0$, $\psi : \mathcal{F}_0 \to \overline{\mathcal{F}}_0$ where $\psi \circ V \circ \phi^{-1} = \pi_r$, $\phi(w(0)) = 0$, $\psi(V_{w(0)}) = 0$ and, for an appropriate choice of bases, $\pi_r$ maps

the coordinates of $\overline{\mathcal{W}}_0$ to the *first* $r$ coordinates of $\overline{\mathcal{F}}_0$, *i.e.*, $(x_1, \ldots, x_p) \mapsto (x_1, \ldots, x_r, 0, 0, \ldots)$, where $r$ is the rank of the operator $DV_{w(0)}$. We denote by $\Pi_r$ the hyperplane in $\mathcal{F}$ spanned by the first $r$ vectors of the basis. We recall that by Abraham et al. (2012); Boutaib (2015); Lee (2003) the maps, $\psi, \phi, \pi_r$ are continuous with Lipschitz derivatives $D\psi, D\phi, D\pi_r$ respectively.

We consider the trajectory of $\overline{V}_{w(t)} := \pi_r \circ \phi(w(t)) = \psi(V_{w(t)})$. Denoting by $D\cdot$ the Fréchet derivative at the corresponding point of the dynamics and noting that $DV = D\psi^{-1} D\pi_r D\phi$ we have

$$
\begin{aligned}
\frac{\mathrm{d}}{\mathrm{d}t} \overline{V}_{w(t)} &= -\frac{1}{\alpha} \langle D\psi DV DV^\top, T^\lambda \alpha \psi^{-1}(\overline{V}_{w(t)}) - \alpha \psi^{-1}(\overline{V}_{w(t)}) \rangle_\pi \\
&= -\frac{1}{\alpha} \langle D\pi_r D\phi D\phi^\top D\pi_r^\top (D\psi^{-1})^\top, T^\lambda \alpha \psi^{-1}(\overline{V}_{w(t)}) - \alpha \psi^{-1}(\overline{V}_{w(t)}) \rangle_\pi, \quad (11)
\end{aligned}
$$

so $\overline{V}$ remains in $\Pi_r$. As a consequence of the above we can naturally define a metric (the pushforward metric) on $\overline{\mathcal{F}}_0$ by the tensor $\bar{g}_{\bar{v}} = (D\pi_r D\phi D\phi^\top D\pi_r^\top)^{-1}$. In fact, by choosing the metric tensor to be constant on $\overline{\mathcal{F}}_0$, *i.e.*, equal to $\bar{g}_0$ for all $v \in \overline{\mathcal{F}}_0$, we equip the linear space $\overline{\mathcal{F}}_0$ with a scalar product $\langle \cdot, \cdot \rangle_0$. This, in turn, directly induces a norm $\| \cdot \|_0$ on the same space. We now proceed to use such simple metric structure to establish the existence and uniqueness of a fixed point of (11) in $\overline{\mathcal{F}}_0$ for $\alpha$ large enough.

The result of our theorem follows from (Simpson-Porco & Bullo, 2014, Proposition 4.1), which establishes uniqueness and exponential contraction at rate $\ell > 0$ of a dynamical system evolving under the flow of a vector field $X$ given by the RHS of (11) in a forward invariant set $\overline{\mathcal{F}}_0$ provided that for every geodesic $\gamma(s)$ in $\overline{\mathcal{F}}_0$ (12) holds. Therefore, the proof of convergence is concluded by applying Lemma 3.6 and Lemma 3.7, whose proofs can be found in supplementary materials. The proof of the optimality of the fixed point is postponed as Lemma A.1 in the appendix. $\qquad\square$

**Lemma 3.6.** *There exists $\delta > 0$ and $\alpha_0 > 0$ such that the ball $\mathcal{B}_\delta^0(0) \subseteq \overline{\mathcal{F}}_0$ is forward invariant and forward complete with respect to the dynamics of (7) for all $\alpha > \alpha_0$.*

**Lemma 3.7.** *There exists $\ell > 0$, $\delta > 0$ and $\alpha_0 > 0$ such that for all $\alpha > \alpha_0$ and all geodesics $\gamma(s)$ contained in the ball $\mathcal{B}_\delta^0(0) \subseteq \overline{\mathcal{F}}_0$, the function*

$$
\langle \gamma'(s), X(\gamma(s)) \rangle_0 - \ell s \langle \gamma'(0), \gamma'(0) \rangle_0, \tag{12}
$$

*is strictly decreasing in $s$.*

**Remark 3.8.** *The proof of Theorem 3.5 can be straightforwardly generalized to the case where the initial condition $V_0$ is not identically $0$ but within $\mathcal{B}_{\varrho(\alpha)}^\mu(0)$ for $\varrho(\alpha)$ going to $0$ with $\alpha \to \infty$. This generalization, however, requires the map $V$ to be* uniformly *Lipschitz smooth for $w \in \mathcal{W}_0$. Among other things, this extension allows to explicitly cover the training of randomly initialized, single layer neural networks.*

## 4 NUMERICAL EXAMPLES

### 4.1 A DIVERGENT NONLINEAR APPROXIMATOR

We illustrate the convergence properties of TD learning in the lazy training regime in the under-parametrized case by applying it to the classical framework of (Tsitsiklis & Van Roy, 1997, Section X). This reference gives an example of a family of nonlinear function approximators that diverge when trained with the TD method. The intuition behind this counterexample is that one can construct a manifold of approximating functions $\mathcal{F}_w$ in the form of a spiral, with the same orientation as the rotation of the vector field induced by the TD update in the space of functions. By choosing the windings of the spiral to be dense enough, the projection of the TD vector field follows the spiral in the outward direction, leading to a divergence of the algorithm, as displayed schematically in Fig. 2a. More specifically, consistently with Tsitsiklis & Van Roy (1997), we parametrize the manifold $\mathcal{F}_w$ as $V_\vartheta := e^{\hat{\varepsilon}\vartheta}(a \cos(\hat{\lambda}\vartheta) - b \sin(\hat{\lambda}\vartheta)) - V^*$ for $a = (10, -7, -3)$, $b = (2.3094, -9.815, 7.5056)$, $\hat{\varepsilon} = 0.01$, $\hat{\lambda} = 0.866$. We choose the discount $\gamma = 0.9$ and a step-size of $\beta_t \equiv 2 \times 10^{-3}$, while the underlying Markov chain is defined by the transition matrix $P_{ij} = (\delta_{j,\mathrm{mod}(i,3)+1} + \delta_{i,j})/2$, where $\delta_{i,j}$ is the Kronecker delta function and equals $1$ if $i = j$ and $0$ else. We note that the step-size does not affect the convergence properties of the algorithm, as argued in Tsitsiklis & Van Roy (1997), where the immediate reward was set to $\bar{r} = (0, 0, 0)$. Note that, as realizing the conditions of Theorem 3.5 would start the simulation at the solution $V^* = (0, 0, 0)$, we shift both the solution and the manifold

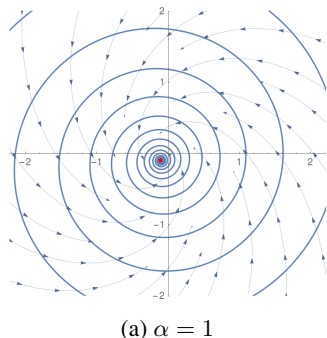

(a) $\alpha = 1$

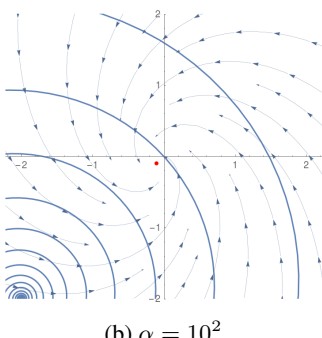

(b) $\alpha = 10^2$

Figure 2: Schematic representation of the manifold $\mathcal{F}_w$ for the example in Section 4.1 before (a) and after (b) scaling of $\alpha$. The underlying vector field represents the TD error $\delta(V)$ from (3), whose projection on $T_\vartheta \mathcal{F}_w$ gives the dynamics of the TD update in $\mathcal{F}_w$. In (a) this projection points "outwards" along the spiral, while (b) it has a fixed point close to 0. The scaling yields an effective "linearization" of the manifold around 0. The red point marks the global fixed point of the vector field.

of approximating functions by the same vector in the embedding space, leaving the new solution $V^* = -V_0 = -a$ at the center of the spiral, *i.e.*, realized at $\vartheta = -\infty$. This corresponds to choosing an average reward $\bar{r} = (-6.85, 8.35, -1.5)$. We note that by the affine nature of the TD update, this change in $\bar{r}$ results in a global shift of the TD vector field in $\mathcal{F}$ and does not affect the update of $\vartheta$. In particular, this means that the TD update remains *divergent* for every initial condition different than the solution $V^*$.

We run the TD update in the off-centered situation both for values of $\alpha = 1$ (the classical, divergent regime) and $\alpha = 100$. As explained in the previous sections, this scaling of the approximating function makes the TD update *convergent*, as displayed in Fig. 3a. Indeed, under this scaling the solution converges to a *local* minimum of the dynamics. The intuition behind the convergence of the algorithm is outlined in Fig. 2: when $\alpha$ is large we are in an almost linear regime where the TD update converges.

## 4.2 SINGLE LAYER NEURAL NETWORKS

We show that the regime of study arises naturally in one hidden layer neural networks for a certain family of initialization. We consider the example of ReLu activation, *i.e.*, when the model is given by

$$V_w(s) = \sum_{i=1}^{N} a_i \max(0, b_i \cdot s - c_i), \tag{13}$$

for $s \in \mathbb{R}^m$ and $N$ distinct $(m+2)$-dimensional vectors $w_i = (a_i, (b_i)_1, \ldots, (b_i)_m, c_i)_{i \in (1,\ldots,N)}$. Typical initialization of the weights of the above model is of the form $a_i \overset{iid}{\sim} \mathcal{N}(0, 1/\sqrt{N})$, $(b_i)_j \overset{iid}{\sim} \mathcal{N}(0, 1/\sqrt{m})$ for all $j$ and $c_i \overset{iid}{\sim} \mathcal{N}(0, 1)$. However, by the linearity of (13) in $a_i$, by the rescaling property of normal distribution this is equivalent to writing

$$\alpha V_w(s) = \alpha \frac{1}{N} \sum_{i=1}^{N} a_i \max(0, b_i \cdot s - c_i), \tag{14}$$

for an $N$-dependent $\alpha(N) = \sqrt{N}$ (diverging in $N$), $a_i \overset{iid}{\sim} \mathcal{N}(0, 1)$, $(b_i)_j \overset{iid}{\sim} \mathcal{N}(0, 1/\sqrt{d})$ and $c_i \overset{iid}{\sim} \mathcal{N}(0, 1)$[1]. Therefore, this common choice of initial conditions implicitly starts the training of the above model in the lazy regime (Ghorbani et al., 2019b). We train the model (14) by TD learning (7) with fixed step-size $\beta_t \equiv 10^{-3}$ both in the over- and under-parametrized regime. To do so, we draw an objective function $V^*$ randomly with distribution $V^*(s) \overset{iid}{\sim} \mathcal{N}(0, 1)$ for all $s \in \mathcal{S}$ on a grid

---

[1]A heuristic justification that the scaling the parameters of the neural network by $\alpha(N)/N = 1/\sqrt{N}$ leads to lazy training while the scaling $N^{-1}$ is natural for the model $V_w$ and does not lead to the lazy regime can be found in Chizat & Bach (2018b). This natural scaling is studied in depth in Chizat & Bach (2018a); Mei et al. (2018); Rotskoff & Vanden-Eijnden (2018)

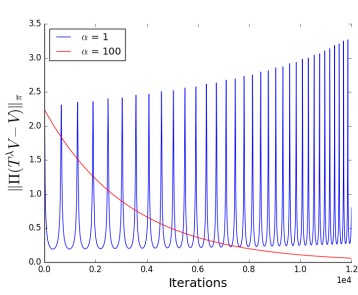

(a) Example from Tsitsiklis & Van Roy (1997)

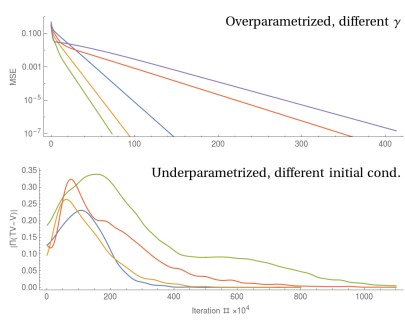

(b) Neural networks simulation

Figure 3: Results of the training of nonlinear value function approximation with TD learning for the examples described in Section 4.1 (a) and Section 4.2 (b). In (a), we plot the $\mu$-norm of the projected TD error $\Pi(T^\lambda V - V)$. This quantity measures the increments of the model parameters during training and vanishes at a local minimum of the TD dynamics. We see that the algorithm diverges for $\alpha = 1$ (blue curve), but converges to a local minimum for $\alpha = 100$. In (b, above) we plot the MSE of single layer neural network during training in the over-parametrized regime ($N = 100, d = 30, \alpha = 500$ ) for different choices of $\gamma$ ($0.8, 0.83, 0.85, 0.87, 0.9$), showing exponential convergence (at different rates) to the global minimum claimed in Theorem 3.1. In (b, below) we again plot the norm of the the projected TD error for a neural network in the under-parametrized regime ($N = 10, d = 50, \alpha = 100$) for different initial conditions, showing that the dynamics converge to a local fixed point.

of $d$ equally spaced points on the interval $[-1, 1]$. We then compute the corresponding average reward by solving the TD equation: $\bar{r} = (\mathbb{1} - \gamma P)V^*$, and train the model (7) for $\lambda = 0$, $\gamma = 0.9$ (when not specified otherwise) with transition matrix $P_{ij} = (\delta_{j,\text{mod}(i,d)+1} + \delta_{i,j})/2$. To respect the conditions of Theorem 3.5, we initialize half of the parameters of the neural network as explained above, while the other half is obtained by replicating the values of $b_i, c_i$ and inverting the one of $a_i \to -a_i$. This "doubling trick" introduced in Chizat & Bach (2018b) produces a neural network with $V_{w(0)} \equiv 0$ and randomly initialized weights with the desired distribution. We consider situations where $N = 10$, $d = 50$ (under-parametrized, taking $\alpha = 100$) and $N = 100$, $d = 30$ (over-parametrized, with $\alpha = 500$), and plot the convergence to local, respectively global minima in Fig. 3b.

## 5 DISCUSSION AND CONCLUSION

In this work we have proven the convergence properties of the TD learning algorithm with nonlinear value function approximation in the lazy training regime. In this regime, the algorithm behaves essentially like a linear approximator spanning the tangential space of the approximating manifold (in function space) at initialization. As such, the training converges exponentially fast with probability one to the global minimum or a local fixed point depending on the codimension of the approximating manifold in the search space. This guarantees convergence with little parametric displacement. This phenomenon can be intuitively understood as an effect of the linearized regime in which the neural networks are trained which reduces them, in the limit, to a randomized kernel method (more precisely a Neural Tangent Kernel (Jacot et al., 2018)). In this sense, convergence of lazy models may come at the expense of their expressivity. Recent works (Chizat & Bach, 2018b; Ghorbani et al., 2019b) discuss the approximating power of lazy neural networks in the supervised setting, highlighting their limits WRT their non-lazy counterparts and naturally comparing them with random feature models (Yehudai & Shamir, 2019), but an exhaustive study of the expressivity of these models, in particular in the context of reinforcement learning is still lacking. Nonetheless, the results proven in this work emphasize the interest of this regime in the framework of deep reinforcement learning, where models often suffer from divergent behavior especially during early stages of training.

Future directions of research include the extension of these results to more complex, nonlinear reinforcement learning algorithms such as Q-learning, and the development of more refined, nonasymptotic versions of the above theorems. Furthermore, a more thorough exploration of the relationship between the limiting results in Chizat & Bach (2018a) and the ones presented here and in Chizat & Bach (2018b) while transposing those to the framework of reinforcement learning would be important for the understanding of the limiting dynamics of neural networks in this domain.

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

## A  SUPPLEMENTARY PROOFS

To simplify the notation in the forthcoming analysis, we slightly abuse the notation used when the state space is finite-dimensional extending it, when necessary, to the infinite-dimensional setting. This naturally generalizes matrix multiplication to the action of linear operators. In particular the action of $\Gamma$, which we recall in the finite-dimensional setting is a diagonal matrix with entries $\mu(s)$, is to be intended as

$$(a^\top \Gamma b)_{ij} = \int_{\mathcal{S}} a_i(s)b_j(s)\,\mu(\mathrm{d}s)\,.$$

Furthermore, we introduce the following decomposition of the TD operator:

$$T^\lambda V = \bar{r}^\lambda + \gamma P^\lambda V\,,$$

where

$$\bar{r}^\lambda(s) := (1-\lambda)\sum_{m=0}^\infty \lambda^m \mathbb{E}_s\left[\sum_{t=0}^m \gamma^t r(s_t,s_{t+1})\right]\,,\qquad P^\lambda V(s) := (1-\lambda)\sum_{m=0}^\infty (\lambda\gamma)^m \mathbb{E}_s\left[V(s_{m+1})\right]\,,$$

or, in vector notation

$$\bar{r}^\lambda := (1-\lambda)\sum_{m=0}^\infty \lambda^m \sum_{t=0}^m \gamma^t P^t r\,,\qquad P^\lambda V(s) := (1-\lambda)\sum_{m=0}^\infty (\lambda\gamma)^m P^{m+1} V\,.$$

In the proofs below, we will use the above, simplified notation to obtain contraction estimates on the dynamical system (4). These estimates will leverage the fact that $P^\lambda$ is nonexpansive and $\gamma < 1$, and from this notation contraction rates in terms of $\gamma$ will arise naturally. However, by Lemma 3.3, we know that the contraction rate of $T^\lambda$ is $\gamma_\lambda$. Rewriting the proofs with $\gamma \to \gamma_\lambda$ will show the stronger contraction.

### A.1  OVER-PARAMETRIZED REGIME

**Lemma 3.2** (Perturbation of the metric). *Let $\mathcal{G}_0$ be a compact subset of a linear space $\mathcal{G}$. For $v(0) \in \mathcal{G}_0$, let $g_v$ be a continuous, self-adjoint linear operator that is positive definite in a neighborhood of $v(0)$ when restricted on $\mathcal{G}$. Then for all $\varepsilon > 0$ there exists $\delta > 0$ such that, for all $v \in \mathcal{B}_\delta(v(0)) \subseteq \mathcal{G}_0$*

$$g_{v(0)} = (\mathbb{1} + \tilde{g}_v)g_v\,,$$

*for a linear operator $\tilde{g}_v : \mathcal{F} \to \mathcal{F}$ with $\|\tilde{g}_v\| < \varepsilon$. More specifically, let $\sigma_{min}$ be the smallest singular value of $DV_{w(0)}$. Then if $\varrho \leq (1-\gamma)\sigma_{min}^2/(48L_{DV})$, (10) holds with $\|\tilde{g}_{V(w)}\| < \frac{1-\gamma}{4}$ for all $w \in \mathcal{B}_\varrho(w(0))$.*

*Proof of Lemma 3.2.* Let $B_w = DV_{w(0)}DV_{w(0)}^\top - DV_w DV_w^\top$. We carry out the proof for the case $\sigma_{\min} < 1$ (else the result holds with $\sigma_{\min} = 1$ in $\varrho$), in which case we have for all $w \in \mathcal{B}_\varrho(w(0))$ that

$$\|B_w\| \leq 2L_{DV}\|w(0) - w\| + (L_{DV}\|w(0) - w\|)^2 \leq 3L_{DV}\|w(0) - w\|\,.$$

Then we can write

$$\begin{aligned}
g_{w(0)} &= (DV_{w(0)}DV_{w(0)}^\top)^{-1} = (DV_w DV_w^\top + B_w)^{-1}\\
&= (g_w^{-1}(1 + g_w B_w))^{-1} = (1 + g_w B_w)^{-1}g_w\\
&= \sum_{n=0}^\infty (-1)^n (g_w B_w)^n g_w = g_w + \sum_{n=1}^\infty (-1)^n (g_w B_w)^n g_w\,.
\end{aligned}$$

Furthermore, by the assumptions on the regularity of $V$ and on the initial condition $w(0)$ we have that $g_w \preceq 4/\sigma_{\min}^2 \mathbb{1}$, provided that $w \in \mathcal{B}_\varrho(w(0))$ for $\varrho$ as in Lemma 3.2. Therefore, the perturbation $\tilde{g}_w := \sum_{n=1}^\infty (-1)^n (g_w B_w)^n$ satisfies

$$\|\tilde{g}_w\| = \|\sum_{n=1}^\infty (-1)^n (g_w B_w)^n\| \leq \sum_{n=1}^\infty \|g_w B_w\|^n \leq \sum_{n=1}^\infty \left(\frac{3L_{DV}}{\sigma_{\min}^2/4}\|w(0) - w\|\right)^n \leq \frac{1-\gamma}{4}\,.$$

The same proof applies in the general case with different, implicit constants.  □

**Lemma 3.3.** *(Tsitsiklis & Van Roy, 1997, Lemmas 1, 3, 7) Under Assumption 1, for any $V, \tilde{V} \in \mathcal{F}$ we have that*

$$\|T^\lambda V - T^\lambda \tilde{V}\|_\mu \le \gamma_\lambda \|V - \tilde{V}\|_\mu \qquad for \quad \gamma_\lambda := \gamma \frac{1-\lambda}{1-\gamma\lambda} \le \gamma < 1. \tag{A.1}$$

*In particular there exists a unique fixed point of $T^\lambda$, $V^* \in \mathcal{F}$ given by (1).*

*Proof of Lemma 3.3.* We first prove that $\|PV\|_\mu \le \|V\|_\mu$. This follows by Jensen inequality and by the invariance of $\mu$:

$$\|PV\|_\mu^2 = V^\top P^\top \Gamma P V = \int_{\mathcal{S}} \mu(\mathrm{d}s) \Big( \int_{\mathcal{S}} P(s, \mathrm{d}s') V(s') \Big)^2$$

$$\le \int_{\mathcal{S}^2} \mu(\mathrm{d}s) P(s, \mathrm{d}s') V(s')^2 = \int_{\mathcal{S}} \mu(\mathrm{d}s) V(s)^2 = \|V\|_\mu^2. \tag{A.2}$$

Then, writing

$$T^\lambda V(s) = (1-\lambda) \sum_{m=0}^\infty \lambda^m \mathbb{E}_s \left[ \sum_{t=0}^m \gamma^t r(s_t, s_{t+1}) + \gamma^{m+1} V(s_{m+1}) \right]$$

$$= (1-\lambda) \sum_{m=0}^\infty \lambda^m \left( \sum_{t=0}^m \gamma^t \mathbb{E}_s [\bar{r}(s_t)] + \gamma^{m+1} \mathbb{E}_s [V(s_{m+1})] \right)$$

$$= (1-\lambda) \sum_{m=0}^\infty \lambda^m \left( \sum_{t=0}^m \gamma^t P^t \bar{r}(s) + (\gamma P)^{m+1} V(s) \right),$$

where $s_t$ is the process on $\mathcal{S}$ induced by $P$ with initial condition $s_0$, we have contraction of the operator $T^\lambda$ in $L^2(\mathcal{S}, \mu)$ by

$$\|T^\lambda(V - \tilde{V})\|_\mu = \left\| (1-\lambda) \sum_{m=0}^\infty \lambda^m (\gamma P)^{m+1} \left( V(s) - \tilde{V}(s) \right) \right\|_\mu$$

$$\le (1-\lambda) \sum_{m=0}^\infty \lambda^m \gamma^{m+1} \left\| V(s) - \tilde{V}(s) \right\|_\mu$$

$$= \frac{\gamma(1-\lambda)}{1-\gamma\lambda} \left\| V(s) - \tilde{V}(s) \right\|_\mu,$$

where in the inequality above we have used (A.2). This proves that $T^\lambda$ is a contraction in $\mathcal{F}$, and as such it must have a unique fixed point. That this fixed point corresponds to (1) is immediately checked by direct computation. $\qquad \square$

**Theorem 3.1** (Over-parametrized case). *Assume that $\sigma_{min} > 0$, where $\sigma_{min}$ is the smallest singular value of $DV_{w(0)}$. Assume further that $w(0)$ is such that $\|V_{w(0)}\|_0 < M := (1-\gamma)^2 \sigma_{min}^2 / (192\kappa^2 L_{DV} \|DV_{w(0)}\|)$, then for $\alpha > \alpha_0 := \|V^*\|_0 / M$ we have for all $t \ge 0$ that*

$$\|V^* - \alpha V_{w(t)}\|_0^2 \le \|V^* - \alpha V_{w(0)}\|_0^2 e^{-\frac{1-\gamma}{2\kappa^2}t}. \tag{A.3}$$

*Recall that $V^*$ is the exact value function given by (1). Moreover, if $\|V_{w(0)}\|_0 \le C\alpha^{-1}$ for a constant $C > 0$, then $\sup_{t>0} \|w(t) - w(0)\| = \mathcal{O}(\alpha^{-1})$.*

*Proof of Theorem 3.1.* By setting $\varrho := (1-\gamma)\sigma_{min}^2 / (48 L_{DV})$ and by the assumed Lipschitz smoothness of $V$, $DV_w \cdot DV_w^\top \succeq \sigma_{min}^2 / 4$ as long as $w \in \mathcal{B}_\varrho(w(0))$. We would like to check a local exponential contraction condition, *i.e.*, that for all $w(t) \in \mathcal{B}_\varrho(w(0))$ we have

$$\frac{\mathrm{d}}{\mathrm{d}t} U(\alpha V_{w(t)}) \le \frac{\gamma-1}{2\kappa^2} U(\alpha V_{w(t)}), \qquad \text{for } t > 0. \tag{A.4}$$

To obtain the above result we apply the chain rule:

$$\frac{\mathrm{d}}{\mathrm{d}t} U(\alpha V_{w(t)}) = \langle \partial_f U(\alpha V_{w(t)}), \frac{\mathrm{d}}{\mathrm{d}t} \alpha V_{w(t)} \rangle_0$$

$$= \alpha \langle \alpha V_{w(t)} - V^*, \, DV_{w(t)} \cdot \frac{\mathrm{d}}{\mathrm{d}\,t} w(t) \rangle_0$$

$$= \langle \alpha V_{w(t)} - V^*, \, DV_{w(t)} \cdot DV_{w(t)}^\top \Gamma (T^\lambda \alpha V_{w(t)} - \alpha V_{w(t)}) \rangle_0. \qquad \text{(A.5)}$$

Throughout, we define $\tau_\varrho := \inf\{t < 0 \, : \, w(t) \notin \mathcal{B}_\varrho(w(0))\}$, $g_w := (DV_w \cdot DV_w^\top)^{-1}$ (recalling that the $DV_w \cdot DV_w^\top$ has full rank in $\mathcal{B}_\varrho(w(0))$) and write $g_0 = (\mathbb{1} + \tilde{g}_w)g_w$, where $\tilde{g}_w$ is defined in Lemma 3.2. Then, as long as $t < \tau_\varrho$ we have, for every $a, b \in \mathcal{F}$

$$\langle a, DV_{w(t)} \cdot DV_{w(t)}^\top \Gamma b \rangle_0 = \langle a, (\mathbb{1} + \tilde{g}_{w(t)})\Gamma b \rangle \leq \langle a, b \rangle_\mu + \|\tilde{g}_{w(t)}\| \|a\|_\mu \|b\|_\mu.$$

By the above result we can bound from above the RHS of (A.5) by

$$\frac{\mathrm{d}}{\mathrm{d}\,t} U(\alpha V_{w(t)}) \leq \langle \alpha V_{w(t)} - V^*, T^\lambda \alpha V_{w(t)} - \alpha V_{w(t)} \rangle_\mu + \|\tilde{g}_{w(t)}\| \|\alpha V_{w(t)} - V^*\|_\mu \|T^\lambda \alpha V_{w(t)} - \alpha V_{w(t)}\|_\mu.$$
$$\text{(A.6)}$$

Recalling that by Lemma 3.3 we have

$$\|T^\lambda \alpha V_{w(t)} - \alpha V_{w(t)}\|_\mu = \|T^\lambda \alpha V_{w(t)} - V^*\|_\mu + \|\alpha V_{w(t)} - V^*\|_\mu \leq 2\|\alpha V_{w(t)} - V^*\|_\mu, \quad \text{(A.7)}$$

and applying Lemma 3.2, we can bound the second term of (A.6) from above as

$$\|\tilde{g}_{w(t)}\| \|\alpha V_{w(t)} - V^*\|_\mu \|T^\lambda \alpha V_{w(t)} - \alpha V_{w(t)}\|_\mu \leq \frac{1 - \gamma}{2} \|\alpha V_{w(t)} - V^*\|_\mu^2. \qquad \text{(A.8)}$$

On the other hand, for the first term we have by Cauchy-Schwartz inequality and (A.1) that

$$\langle \alpha V_{w(t)} - V^*, T^\lambda \alpha V_{w(t)} - \alpha V_{w(t)} \rangle_\mu = \langle \alpha V_{w(t)} - V^*, (T^\lambda \alpha V_{w(t)} - V^*) - (\alpha V_{w(t)} - V^*) \rangle_\mu,$$
$$\leq \|\alpha V_{w(t)} - V^*\|_\mu \|T^\lambda \alpha V_{w(t)} - V^*\|_\mu - \|\alpha V_{w(t)} - V^*\|_\mu^2$$
$$\leq (\gamma - 1)\|\alpha V_{w(t)} - V^*\|_\mu^2, \qquad \text{(A.9)}$$

where $\gamma$ is the contraction rate of the TD difference in $\mathcal{F}$, see (A.1). Finally, combining (A.8) and (A.9) we obtain

$$\frac{\mathrm{d}}{\mathrm{d}\,t} U(\alpha V_{w(t)}) \leq \frac{\gamma - 1}{2} \|\alpha V_{w(t)} - V^*\|_\mu^2 \leq \frac{\gamma - 1}{2\kappa^2} \|\alpha V_{w(t)} - V^*\|_0^2, \qquad \text{(A.10)}$$

and the last inequality results from the equivalence of norms $\|\cdot\|_0$ and $\|\cdot\|_\mu$ (both have full support on a finite set). The desired result (A.3) follows directly from the above by Grönwall's inequality for all $t < \tau_\varrho$.

It now only remains to show that under the given choice of $\alpha$, we have $\tau_\varrho = \infty$. By the contraction of $T^\lambda$ Lemma 3.3 and our choice of $\varrho < \sigma_{\min}/(2L_{DV})$ we write

$$\left\| \frac{\mathrm{d}}{\mathrm{d}\,t} w(t) \right\|_2 \leq \frac{1}{\alpha} \|DV_{w(t)}\| \|T^\lambda \alpha V_{w(t)} - \alpha V_{w(t)}\|_\mu \leq \frac{2}{\alpha} \|DV_{w(0)}\| \|\alpha V_{w(t)} - V^*\|_\mu.$$

Integrating the above and combining with the result from (A.10) in the previous paragraph we have

$$\|w(t) - w(0)\|_2 \leq \frac{2}{\alpha} \|DV_{w(0)}\| \|\alpha V_{w(0)} - V^*\|_0 \int_0^t \exp\left[ \frac{\gamma - 1}{2\kappa^2} s \right] \mathrm{d}s$$

$$\leq \frac{4\kappa^2}{\alpha(1 - \gamma)} \|DV_{w(0)}\| \|\alpha V_{w(0)} - V^*\|_0. \qquad \text{(A.11)}$$

Given that $\|\alpha V_{w(0)} - V^*\|_0 \leq 2\alpha M$, the above quantity is bounded by $\varrho$ and therefore $\tau_\varrho = \infty$, as desired.

Finally, from (A.11) we see that if $\|V_{w(0)}\|_0 \leq C\alpha^{-1}$ then $\|w(t) - w(0)\|_2 \leq \frac{4\kappa^2}{\alpha(1-\gamma)} \|DV_{w(0)}\|(C + M\alpha_0) = \mathcal{O}(\alpha^{-1})$ for all $t > 0$. $\qquad \square$

## A.2 Under-parametrized regime

**Lemma 3.6.** *There exists $\delta > 0$ and $\alpha_0 > 0$ such that the ball $\mathcal{B}_\delta^0(0) \subseteq \overline{\mathcal{F}}_0$ is forward invariant and forward complete with respect to the dynamics of (7) for all $\alpha > \alpha_0$.*

*Proof of Lemma 3.6.* We define the Lyapunov function $\bar{U}(f) := \frac{1}{2}\|f\|_0^2$, whose sublevel sets are $\mathcal{B}_\delta^0(0)$. We prove forward invariance of such sets by showing that, on their boundary (*i.e.*, on the sphere $S_\delta^{r-1} \subset \bar{\mathcal{F}}_0$ of radius $\delta$), $\bar{U}(f)$ decreases along trajectories of (7) for $\alpha$ large enough. Noting that $S_\delta^{r-1} \subset \overline{\mathcal{F}}_0$ upon taking $\delta$ small enough, we differentiate $\bar{U}(\overline{V}_{w(t)})$ WRT time for $w(t)$ obeying (7) at points $\overline{V} := \overline{V}_{w(t)} \in S_\delta^{r-1}$:

$$
\begin{aligned}
\frac{\mathrm{d}}{\mathrm{d}t}\bar{U}(\overline{V}) &= \frac{1}{\alpha}\langle \overline{V}, \bar{g}_{w(t)}^{-1} D\psi_{\overline{V}}^{-1}\Gamma(T^\lambda \alpha \psi^{-1}(\overline{V}) - \alpha\psi^{-1}(\overline{V}))\rangle_0 \\
&= \frac{1}{\alpha}\langle \overline{V}, (D\psi_{\overline{V}}^{-1})^\top \Gamma(T^\lambda \alpha\psi^{-1}(\overline{V}) - \alpha\psi^{-1}(\overline{V}))\rangle + R_g(\overline{V}) \\
&= \frac{1}{\alpha}\langle D\psi_{\overline{V}}^{-1}\overline{V}, \bar{r}^\lambda + \alpha(\gamma P^\lambda - \mathbb{1})\psi^{-1}(\overline{V})\rangle_\mu + R_g(\overline{V}) \\
&\leq \langle D\psi_{\overline{V}}^{-1}\overline{V}, (\gamma P^\lambda - \mathbb{1})\psi^{-1}(\overline{V})\rangle_\mu + \frac{1}{\alpha}\|D\psi_{\overline{V}}^{-1}\overline{V}\|_\mu\|\bar{r}^\lambda\|_\mu + |R_g(\overline{V})| . \quad \text{(A.12)}
\end{aligned}
$$

where we have defined $R_g(\overline{V}) := \frac{1}{\alpha}\langle \overline{V}, \tilde{g}_{w(t)}(D\psi_{\overline{V}}^{-1})^\top \Gamma(T^\lambda \alpha\psi^{-1}(\overline{V}) - \alpha\psi^{-1}(\overline{V}))\rangle$ for $\tilde{g}_w$ from Lemma 3.2. We now proceed to bound the last two terms on the RHS from above. The second term is of order $\alpha^{-1}$ and therefore goes to 0 for $\alpha \to \infty$ while for the last one we have that, by the equivalence of the norms $\| \cdot \|_\mu$ and $\| \cdot \|_2$,

$$
\begin{aligned}
|R_g(\overline{V})| &\leq \frac{1}{\alpha}\|\overline{V}\|_2\|\tilde{g}_{w(t)}\|\|(D\psi_{\overline{V}}^{-1})^\top \Gamma\left[\bar{r}^\lambda + (\gamma P^\lambda - \mathbb{1})\alpha\psi^{-1}(\overline{V})\right]\|_2 \\
&\leq \frac{1}{\alpha}\|\overline{V}\|_2\|\tilde{g}_{w(t)}\|\|(D\psi_{\overline{V}}^{-1})^\top \Gamma\bar{r}^\lambda\| + \|\overline{V}\|_2\|\tilde{g}_{w(t)}\|\|(D\psi_{\overline{V}}^{-1})^\top\Gamma(\gamma P^\lambda - \mathbb{1})\psi^{-1}(\overline{V})\|_2 \\
&\leq \alpha^{-1}C + \varepsilon_R(\delta)\|\overline{V}\|_\mu^2 . \quad \text{(A.13)}
\end{aligned}
$$

for a constant $C$ bounded by the norm of all operators and, by Lemma 3.2 a positive function $\varepsilon_R(\delta)$ with $\lim_{\delta\to 0}\varepsilon_R(\delta) = 0$. By the bounds established above and the fact that $\|\overline{V}\|_\mu \geq \kappa^{-1}\delta$ for $\overline{V} \in S_\delta^{r-1} \subset \overline{\mathcal{F}}_0$ it is sufficient to show that the first term in (A.12) satisfies

$$
\langle D\psi_{\overline{V}}^{-1}\overline{V}, (\gamma P^\lambda - \mathbb{1})\psi^{-1}(\overline{V})\rangle_\mu \leq -\varepsilon\|\overline{V}\|_\mu^2 , \quad \text{(A.14)}
$$

for $\delta$ small enough and a constant $\varepsilon > 0$ independent of $\delta$. We Taylor-expand $\psi^{-1}$ around the origin, denoting the second order remainder of that expansion by $R_2(\cdot, \cdot)$, and since $\psi^{-1}(\overline{V}_0) = 0$ we have,

$$
\begin{aligned}
\langle D\psi_{\overline{V}}^{-1}\overline{V}, (\gamma P^\lambda - \mathbb{1})\psi^{-1}(\overline{V})\rangle_\mu &= \langle D\psi_{\overline{V}}^{-1}\overline{V}, (\gamma P^\lambda - \mathbb{1})D\psi_0^{-1}\overline{V}\rangle_\mu \\
&\quad + \langle D\psi_{\overline{V}}^{-1}\overline{V}, (\gamma P^\lambda - \mathbb{1})R_2(\overline{V}, \overline{V})\rangle_\mu , \quad \text{(A.15)}
\end{aligned}
$$

where we have introduced the short hand notation $D\psi_0^{-1} = D\psi_{\overline{V}_0}^{-1}$. By the Lipschitz smoothness of $\psi^{-1}(\cdot)$ (Lee, 2003) we can bound the norm of the second term from above as

$$
\langle D\psi_{\overline{V}}^{-1}\overline{V}, (\gamma P^\lambda - \mathbb{1})R_2(\overline{V}, \overline{V})\rangle_\mu \leq 2\|D\psi_{\overline{V}}^{-1}\overline{V}\|_\mu\|R_2(\overline{V}, \overline{V})\|_\mu \leq 2L_{D\psi^{-1}}\|D\psi_{\overline{V}}^{-1}\|\|\overline{V}\|_\mu^3 . \quad \text{(A.16)}
$$

For the first term in (A.15) we can also expand $D\psi_{\overline{V}}^{-1} = D\psi_0^{-1} + \tilde{R}_2(\overline{V}, \cdot)$, and by applying a similar bound as (A.16) we obtain that

$$
\langle D\psi_{\overline{V}}^{-1}\overline{V}, (\gamma P^\lambda - \mathbb{1})D\psi_0^{-1}\overline{V}\rangle_\mu \leq \langle D\psi_0^{-1}\overline{V}, (\gamma P^\lambda - \mathbb{1})D\psi_0^{-1}\overline{V}\rangle_\mu + 2L_{D\psi^{-1}}\|D\psi_0^{-1}\|\|\overline{V}\|_\mu^3 . \quad \text{(A.17)}
$$

The second term of the above equation being $\mathcal{O}(\|\overline{V}\|^3)$, we now consider the first one. By the nonexpansion of $P$ in $\| \cdot \|_\mu$ proven in Lemma 3.3 we have

$$
\begin{aligned}
\langle D\psi_0^{-1}\overline{V}, (\gamma P^\lambda - \mathbb{1})D\psi_0^{-1}\overline{V}\rangle_\mu &\leq \gamma\|D\psi_0^{-1}\overline{V}\|_\mu\|P^\lambda D\psi_0^{-1}\overline{V}\|_\mu - \|D\psi_0^{-1}\overline{V}\|_\mu^2 \\
&\leq (\gamma - 1)\|D\psi_0^{-1}\overline{V}\|_\mu^2 \leq (\gamma - 1)(\sigma_{\min}^{D\psi^{-1}})^2\|\overline{V}\|_\mu^2 , \quad \text{(A.18)}
\end{aligned}
$$

where $\sigma_{\min}^{D\psi^{-1}}$ denotes the smallest singular value of $D\psi^{-1}$ in $\mathcal{B}_\delta^0(0)$. Combining (A.16), (A.17) and (A.18) we finally obtain

$$
\langle D\psi_{\overline{V}}^{-1}\overline{V}, (\gamma P^\lambda - \mathbb{1})\psi^{-1}(\overline{V})\rangle_\mu \leq \|\overline{V}\|_\mu^2((\gamma - 1)(\sigma_{\min}^{D\psi^{-1}})^2 + C'\kappa^{-1}\|\overline{V}\|_0) , \quad \text{(A.19)}
$$

for $C' = 2L_{D\psi^{-1}}(\|D\psi_0^{-1}\| + \|D\psi_{\overline{V}}^{-1}\|)$ and recalling that $\kappa$ is the equivalence constant between the norms $\|\cdot\|_\mu$ and $\|\cdot\|_0$ in $\overline{\mathcal{F}}_0$. [2] Therefore, choosing $\delta$ small enough we obtain (A.14) and conclude the proof of forward invariance.

By boundedness of $\mathcal{B}_\delta^0(0)$ in $\overline{\mathcal{F}}_0$, forward completeness follows directly from forward invariance. $\qquad\square$

**Lemma 3.7.** *There exists $\ell > 0$, $\delta > 0$ and $\alpha_0 > 0$ such that for all $\alpha > \alpha_0$ and all geodesics $\gamma(s)$ contained in the ball $\mathcal{B}_\delta^0(0) \subseteq \overline{\mathcal{F}}_0$, the function*

$$\langle \gamma'(s), X(\gamma(s)) \rangle_0 - \ell s \langle \gamma'(0), \gamma'(0) \rangle_0 \,,$$

*is strictly decreasing in $s$.*

*Proof of Lemma 3.7.* To simplify the notation and the forthcoming computation, we prove the differential version of the desired result, *i.e.*, we show that there exists $\ell > 0$ such that

$$\frac{\mathrm{d}}{\mathrm{d}s} \left[ \langle \gamma'(s), X(\gamma(s)) \rangle_0 - \ell s \langle \gamma'(0), \gamma'(0) \rangle_0 \right] < 0 \,. \tag{A.20}$$

The above expression exists almost everywhere by Lipschitz continuity of the terms to be differentiated. When this is not the case, we must interpret this derivative in the sense of distributions. We will highlight the steps where this could be necessary as we go along the proof.

In our case, $X$ is the RHS of (11) mapped through $\psi$ onto $\overline{\mathcal{F}}_0$, *i.e.*,

$$X(\gamma(s)) = -\frac{1}{\alpha} \bar{g}_{\gamma(s)}^{-1} (D\psi_{\gamma(s)}^{-1})^\top \Gamma(T^\lambda \alpha \psi^{-1}(\gamma(s)) - \alpha \psi^{-1}(\gamma(s))) \,.$$

We are going to consider the "flattened" manifold obtained by the maps $\phi$ and $\psi$ equipped with the metric $\bar{g}_0$. In this space, geodesics have the form $\gamma(s) = v_1 + s\Delta v$ where $\Delta v := v_2 - v_1$ for $v_1, v_2 \in \overline{\mathcal{F}}_0$ and their derivative is $\gamma'(s) = \Delta v$. Consequently (A.20) reads

$$\langle \Delta v, \frac{\mathrm{d}}{\mathrm{d}s} X(\gamma(s)) \rangle_0 < \ell \|\Delta v\|_0^2 \,, \tag{A.21}$$

where defining $\tilde{g}_{\gamma(s)} := \bar{g}_0 \bar{g}_{\gamma(s)}^{-1} - \mathbb{1}$ as in Lemma 3.2 we have

$$\frac{\mathrm{d}}{\mathrm{d}s} X(\gamma(s)) = \frac{\mathrm{d}}{\mathrm{d}s} \bar{g}_0 \bar{g}_{\gamma(s)}^{-1} (D\psi_{\gamma(s)}^{-1})^\top \Gamma(T^\lambda(\alpha \psi^{-1}(\gamma(s))) - \alpha \psi^{-1}(\gamma(s)))$$

$$= \frac{\mathrm{d}}{\mathrm{d}s} \bar{X}(\gamma(s)) + \tilde{g}_{\gamma(s)} \frac{\mathrm{d}}{\mathrm{d}s} \bar{X}(\gamma(s)) + D\tilde{g}_{\gamma(s)}(\bar{X}(\gamma(s)), \gamma'(s)) \,. \tag{A.22}$$

for

$$\bar{X}(\gamma(s)) := (D\psi_{\gamma(s)}^{-1})^\top \Gamma(T^\lambda(\alpha \psi^{-1}(\gamma(s))) - \alpha \psi^{-1}(\gamma(s))) \,.$$

We proceed by analyzing the first term in the above equation and leave the task of bounding the last two for later. Using $\partial_s \alpha \psi^{-1}(\gamma(s)) = \alpha D\psi_{\gamma(s)}^{-1} \gamma'(s) = \alpha D\psi_{\gamma(s)}^{-1} \Delta v$ we have that

$$\frac{\mathrm{d}}{\mathrm{d}s} \bar{X}(\gamma(s)) = \frac{1}{\alpha} (D^2 \psi_{\gamma(s)}^{-1})^\top (\Gamma(T^\lambda \alpha \psi^{-1}(\gamma(s)) - \alpha \psi^{-1}(\gamma(s))), \Delta v) \tag{A.23}$$

$$+ (D\psi_{\gamma(s)}^{-1})^\top \Gamma \left[ DT^\lambda D\psi_{\gamma(s)}^{-1} \Delta v - D\psi_{\gamma(s)}^{-1} \Delta v \right] ,$$

where $(D^2 \psi_{\gamma(s)}^{-1})^\top$ denotes the inversion of the last two indices of the Hessian. We now proceed to consider the two terms in the sum above separately (multiplied by the scalar product of (A.21)), defining throughout $(TD)_s := \Gamma(T^\lambda \alpha \psi^{-1}(\gamma(s)) - \alpha \psi^{-1}(\gamma(s)))$. For the first term we have:

$$\frac{1}{\alpha} \langle \Delta v, D^2 \psi_{\gamma(s)}^{-1} (TD_s, \Delta v) \rangle_0 \leq \|\Delta v\|_0^2 \|D^2 \psi_{\gamma(s)}^{-1} \left( \alpha^{-1} \bar{r}^\lambda + (\gamma P^\lambda - \mathbb{1}) \psi^{-1} \gamma(s) \right) \| \leq \varepsilon' \|\Delta v\|_0^2 \,, \tag{A.24}$$

---

[2] We recall that by the construction of the mappsings $\psi, \phi, \pi_r$ and by our assumption in Theorem 3.5 the metric tensor $\bar{g}_t$ has full rank on $\mathcal{F}_0$ and being the latter set compact its eigenvalues are uniformly bounded from below. At the same time, we can equip $\overline{\mathcal{F}}_0$ with the metric induced by $\Gamma$ by restricting it to its first $r$ elements, which are uniformly bounded from below. Hence, the two metrics are equivalent on this space for some equivalence constant $\kappa$.

for any $\varepsilon' > 0$ by using the linearity of the Hessian and bounding its operator norm of $\psi^{-1}$ on a compact space in $\mathcal{F}_0$ while choosing $\alpha$ large enough and $\delta$ small enough, since $\gamma(s) \in \mathcal{B}_\delta^0(0)$. Note that if $D\psi^{-1}$ is not differentiable, the above computation is to be understood in the sense of distributions.

We now focus on the second term of (A.23). In this case we incorporate the operator $\Gamma$ in the inner product and write this term as

$$\langle D\psi_{\gamma(s)}^{-1}\Delta v, DT^\lambda D\psi_{\gamma(s)}^{-1}\Delta v\rangle_\mu - \|D\psi_{\gamma(s)}^{-1}\Delta v\|_\mu^2.$$

Now, by the contraction property of $T^\lambda$ onto the tangential space $T_{\psi_{\gamma(s)}^{-1}}\mathcal{F}$ in the norm $\|\cdot\|_\mu$ we can write

$$\langle D\psi_{\gamma(s)}^{-1}\Delta v, DT^\lambda D\psi_{\gamma(s)}^{-1}\Delta v\rangle_\mu \leq \|D\psi_{\gamma(s)}^{-1}\Delta v\|_\mu \|P^\lambda D\psi_{\gamma(s)}^{-1}\Delta v\|_\mu \leq \gamma\|D\psi_{\gamma(s)}^{-1}\Delta v\|_\mu^2,$$

so that

$$\langle D\psi_{\gamma(s)}^{-1}\Delta v, DT^\lambda D\psi_{\gamma(s)}^{-1}\Delta v\rangle_\mu - \|D\psi_{\gamma(s)}^{-1}\Delta v\|_\mu^2 \leq (\gamma - 1)\|D\psi_{\gamma(s)}^{-1}\Delta v\|_\mu^2. \tag{A.25}$$

Denoting by $\sigma_{\max}^{D\psi^{-1}}, \sigma_{\min}^{D\psi^{-1}}$ the largest and smallest, respectively, singular values of the map $D\psi^{-1}$ in $\mathcal{B}_\delta^0(0)$ (which are bounded away from 0 upon possibly making this set smaller), by nondegeneracy of $D\psi^{-1}$ and by the equivalence of the $\|\cdot\|_\mu$ and $\|\cdot\|_0$ norms on $\overline{\mathcal{F}}_0$ we have that

$$\kappa^{-1}\sigma_{\min}^{D\psi^{-1}}\|\Delta v\|_0 \leq \|\Delta v\|_\mu \sigma_{\min}^{D\psi^{-1}} \leq \|D\psi_{\gamma(s)}^{-1}\Delta v\|_\mu \leq \|\Delta v\|_\mu \sigma_{\max}^{D\psi^{-1}} \leq \kappa\|\Delta v\|_0 \sigma_{\max}^{D\psi^{-1}}.$$

Thus we have

$$\|D\psi_{\gamma(s)}^{-1}\Delta v\|_\mu^2 \geq \kappa^{-2}\left(\sigma_{\min}^{D\psi^{-1}}\right)^2\|\Delta v\|_0^2. \tag{A.26}$$

Getting back to the last two terms in (A.12), we immediately see from Lemma 3.2 that $\tilde{g}_{\gamma(s)}$ is a small, Lipschitz continuous perturbation. Hence, the product

$$\langle \gamma'(s), \tilde{g}_{\gamma(s)}\bar{X}'(\gamma(s))\rangle$$

can be bounded from above similarly to (A.13), while the second order derivative in the third term of (A.22) can be dealt with analogously to what is done in (A.24), giving terms $\varepsilon''\|\Delta v\|_0^2$ and $\varepsilon^{(3)}\|\Delta v\|_0^2$ respectively, both going to 0 as $\delta \to 0$.

Therefore, combining the above with (A.24), (A.25) and (A.26) we have

$$\langle \Delta v, \frac{d}{dt}\bar{X}(\gamma(s))\rangle_0 \leq \frac{\gamma - 1}{\kappa^2}\left(\sigma_{\min}^{D\psi^{-1}}\right)^2\|\Delta v\|_0^2 + \left(\sum_i^3 \varepsilon^{(i)}(\delta)\right)\|\Delta v\|_0^2$$

$$\leq \frac{\gamma - 1}{2\kappa^2}\left(\sigma_{\min}^{D\psi^{-1}}\right)^2\|\Delta v\|_0^2.$$

This directly gives (A.21) by choosing $\ell$ large enough. $\qquad\square$

The next lemma estimates the distance between the fixed point $\tilde{V}^*$ of the dynamics (7) and $V^*$ given by (1), showing that it is close, for large values of $\alpha$ to the best linear model in the tangent space of $\mathcal{F}_w$ at $V_{w(0)}$, given by $\Pi_0 V^*$. We recall that the projection operator $\Pi_0$ onto the linear space spanned by the columns of $DV$ is given by (Tsitsiklis & Van Roy, 1997, Eq. (1))

$$\Pi_0 W := \underset{\{DV_{w(0)}\Delta w \,:\, \Delta w \in \mathbb{R}^p\}}{\arg\min} \|DV_{w(0)}\Delta w - W\|_\mu = DV_{w(0)}(DV_{w(0)}^\top \Gamma DV_{w(0)})^{-1}DV_{w(0)}^\top \Gamma W,$$

for all $W \in \mathcal{F}$ where, if necessary, we interpret $(DV_{w(0)}^\top \Gamma DV_{w(0)})^{-1}$ as a pseudo-inverse.

**Lemma A.1.** *Let $\tilde{V}^*$ be the fixed point of (7) and $V^*$ be the global fixed point of the TD operator, given by (1). Then under the assumptions of Theorem 3.5 there exists constants $\alpha_0 > 0$ and $C^* > 0$ (independent of $\alpha_0$), such that*

$$\|\tilde{V}^* - V^*\|_\mu < \frac{1 - \lambda\gamma}{1 - \gamma}\|\Pi_0 V^* - V^*\|_\mu + C^*\alpha^{-1}, \tag{A.27}$$

*where $\Pi_0$ is the projection operator onto $T_{V(w(0))}\mathcal{F}_w$.*

To prove the above result we compare the dynamics (7) to the dynamics of the model $V$ when *linearized* at $w(0)$. In this case, the dynamics of the parameters is given by

$$\frac{\mathrm{d}}{\mathrm{d}\, t}\bar{w}(t) = DV_{w(0)}^\top \Gamma(T^\lambda \mathcal{V}_{\bar{w}(t)} - \mathcal{V}_{\bar{w}(t)})\,, \tag{A.28}$$

where $\mathcal{V} \in \mathcal{F}$ is the linear, tangent model of $V$ at $w(0)$ defined as

$$\mathcal{V}_w := V_{w(0)} + DV_{w(0)}(w - w(0))\,. \tag{A.29}$$

We can also write the dynamics of the linear model as

$$\frac{\mathrm{d}}{\mathrm{d}\, t}\mathcal{V}_{\bar{w}(t)} := DV_{w(0)} \cdot DV_{w(0)}^\top \Gamma(T^\lambda \mathcal{V}_{\bar{w}(t)} - \mathcal{V}_{\bar{w}(t)})\,. \tag{A.30}$$

Scaling the model as $\mathcal{V} \to \alpha\mathcal{V}$ and $t \to \alpha^{-1}t$ we obtain the analogue of (7):

$$\frac{\mathrm{d}}{\mathrm{d}\, t}\bar{w}(t) := \frac{1}{\alpha}DV_{w(0)}^\top \Gamma(T^\lambda \alpha\mathcal{V}_{\bar{w}(t)} - \alpha\mathcal{V}_{\bar{w}(t)})\,. \tag{A.31}$$

which in $\mathcal{F}$ reads

$$\frac{\mathrm{d}}{\mathrm{d}\, t}\alpha\mathcal{V}_{\bar{w}(t)} := DV_{w(0)} \cdot DV_{w(0)}^\top \Gamma(T^\lambda \alpha\mathcal{V}_{\bar{w}(t)} - \alpha\mathcal{V}_{\bar{w}(t)})\,.$$

*Proof of Lemma A.1.* Recall from (Tsitsiklis & Van Roy, 1997, Lemma 6) that for the linear value function approximation one has

$$\|\mathcal{V}^* - V^*\|_\mu < \frac{1 - \lambda\gamma}{1 - \gamma}\|\Pi_0 V^* - V^*\|_\mu\,, \tag{A.32}$$

where $\Pi_0$ is the projection on $T_{V(w(0))}\mathcal{F}_w$ and $\mathcal{V}^*$ is the unique fixed point of the dynamics (A.30) on that space. In light of this result, our task reduces to bounding the distance between the trajectories of the original (*i.e.*, dynamics (7)) and the linearized model (*i.e.*, dynamics (A.31)) by $C\alpha^{-1}$ for $C$ large enough. We do so in 3 main steps. First of all, we bound the maximal excursion of the models $\mathcal{V}$ and $V$. Mapping both dynamics onto a common coordinate space, we then bound from above the distance between the two trajectories in this space by $\mathcal{O}(\alpha^{-1})$. Finally, we map the dynamics back to the embedding space and show that the correction is again of the same order $\mathcal{O}(\alpha^{-1})$.

**Bounding the maximal excursion.** To compare the dynamics of $\alpha V_{w(t)}$ and $\alpha\mathcal{V}_{\bar{w}(t)}$ we map them to a common space. Recalling the definition of the maps $\phi, \pi_r, \psi$ from the proof of Theorem 3.5 we note that the first order expansion of $\psi$, maps $T_{V(w(0))}\mathcal{F}_w$ to $\overline{\mathcal{F}}_0$. Explicitly, for $\overline{V} \in \overline{\mathcal{F}}_0$ and for $\Delta\mathcal{V} \in T_{V(w(0))}\mathcal{F}_w$ with $\|\Delta\mathcal{V}\|_0$ small enough we have

$$\bar{\psi}(V_{w(0)} + \Delta\mathcal{V}) := D\psi_0\Delta\mathcal{V} \qquad \text{and} \qquad \bar{\psi}^{-1}(\overline{V}) = V_{w(0)} + D\psi_0^{-1}\overline{V} \in T_{V(w(0))}\mathcal{F}_w\,. \tag{A.33}$$

Now, we proceed to show that the dynamics of (7) and (A.31), mapped to $\mathcal{F}_0$, do not exit a ball $\mathcal{B}_\delta^0(0)$, when choosing $\delta = C/\alpha$ for $C$ large enough. We show this with the same strategy used for the proof of Lemma 3.6, *i.e.*, we show that $\bar{U}(f) := \frac{1}{2}\|f\|_0^2$ decreases on $S_\delta^{r-1}(0)$ along the trajectories of interest (note that $\delta$ is now much smaller than that used in Lemma 3.6). We will start with the curved dynamics (7) and will then show that the same result follows, in a simpler setting, for (A.31). For $\overline{V} := \overline{V}_{w(t)} \in S_\delta^{r-1}(0)$ we start by bounding, as in (A.12), the derivative

$$\frac{\mathrm{d}}{\mathrm{d}\, t}\bar{U}(\overline{V}) \leq \langle D\psi_{\overline{V}}^{-1}\overline{V}, (\gamma P^\lambda - \mathbb{1})\psi^{-1}(\overline{V})\rangle_\mu + \frac{1}{\alpha}\|D\psi_{\overline{V}}^{-1}\overline{V}\|_\mu\|\bar{r}^\lambda\|_\mu + |R_g(\overline{V})|\,. \tag{A.34}$$

Before bounding the above terms we recall that by Lipschitz smoothness of $\psi$ we have that

$$\|\psi^{-1}(\overline{V})\| < \|V_{w(0)}\| + \|D\psi_0^{-1}\overline{V}\| + L_{D\psi^{-1}}\|\overline{V}\|^2\,. \tag{A.35}$$

Then, since $V_{w(0)} = 0$, similarly to (A.12) we have for the last term in (A.34) that, for $\alpha$ large enough,

$$|R_g(\overline{V})| \leq \|\tilde{g}_w\|\|\overline{V}\|_2\Big(\|\overline{V}\|_2\|(D\psi_{\overline{V}}^{-1})^\top\Gamma(\gamma P^\lambda - \mathbb{1})\|(\|D\psi_0^{-1}\| + L_{DV}\|\overline{V}\|_2)$$
$$+ \frac{1}{\alpha}\|(D\psi_{\overline{V}}^{-1})^\top\Gamma\bar{r}^\lambda\|_2\Big)\,.$$

By the equivalence of the norms $\|\cdot\|_\mu$, $\|\cdot\|_2$ and $\|\cdot\|_0$ on $\Pi_r$ and since $\delta = C/\alpha$ we have that

$$|R_g(\overline{V})| \le \|\tilde{g}_w\| \|\overline{V}\|_0^2 (K+1) + \mathcal{O}(\alpha^{-3}), \tag{A.36}$$

upon increasing $C$ if necessary and defining $K = \kappa_2^2 \|(D\psi_{\overline{V}}^{-1})^\top \Gamma(\gamma P^\lambda - \mathbb{1})\| \|D\psi_0^{-1}\|$ for $\kappa_2$ the equivalence constant between $\|\cdot\|_2$ and $\|\cdot\|_0$ on $\Pi_r$. The second term in (A.34) can be bounded similarly to the above by the equivalence of norms:

$$\frac{1}{\alpha} \|D\psi_{\overline{V}}^{-1}\overline{V}\|_\mu \|\bar{r}^\lambda\|_\mu \le \|\overline{V}\|_0^2 \frac{\kappa^2 \|D\psi_{\overline{V}}^{-1}\| \|\bar{r}^\lambda\|_\mu}{C}. \tag{A.37}$$

The first term in (A.34) can be treated identically to the proof of Lemma 3.6 to obtain (A.19). Changing the norm in (A.19) and combining it with (A.36) and (A.37) gives

$$\frac{\mathrm{d}}{\mathrm{d}t}\bar{U}(\overline{V}) \le \|\overline{V}\|_0^2 \left( \frac{\gamma-1}{2\kappa^2}(\sigma_{\min}^{D\psi^{-1}})^2 + \frac{\kappa^2 \|D\psi_{\overline{V}}^{-1}\| \|\bar{r}^\lambda\|_\mu}{C} + \|\tilde{g}_w\|(K+1) \right) + \mathcal{O}(\alpha^{-3}).$$

Since $\gamma - 1 < 0$, we can choose $C$ large enough to make the second term in brackets smaller than $(\gamma-1)/12\kappa^2(\sigma_{\min}^{D\psi^{-1}})^2$. The same holds for the third term in brackets by (10), and for the higher order term by taking $\alpha$ large enough, showing that

$$\frac{\mathrm{d}}{\mathrm{d}t}\bar{U}(\overline{V}) \le \frac{\gamma-1}{4\kappa^2}(\sigma_{\min}^{D\psi^{-1}})^2 \|\overline{V}\|_0^2 < 0,$$

as desired. We note that the same reasoning with $L_{DV} = 0$ and $D\psi_{\overline{V}}^{-1} \equiv D\psi_0^{-1}$ yields an identical conclusion for the dynamics of $\mathcal{V}$ in a ball of radius $\delta = C/\alpha$ for $C, \alpha$ large enough. Also, we note that combining the above computation with (A.16) yields

$$\|D\psi_{\overline{V}}^{-1}\Gamma(T^\lambda \alpha\psi^{-1}(\overline{V}) - \alpha\psi^{-1}(\overline{V}))\| \le \|D\psi_{\overline{V}}^{-1}\Gamma\|(\|\bar{r}^\lambda\| + \alpha(\gamma+1)\|D\psi_0^{-1}\overline{V}\| + \alpha L_{D\psi_{\overline{V}}^{-1}}\|\overline{V}\|^2)$$

$$\le (\gamma+1)\|D\psi_{\overline{V}}^{-1}\Gamma\|(\|D\psi_0^{-1}\|C + \|\bar{r}^\lambda\| + \mathcal{O}(\alpha^{-1}))$$

$$\le C_0, \tag{A.38}$$

for $C_0$ large enough, where $D\psi_{\overline{V}}^{-1}\Gamma$ is considered as an operator mapping $\mathcal{F}_0 \to \bar{\mathcal{F}}_0$.

**Bounding the distance of trajectories.** The distance between two trajectories with the same initial condition can be bounded by $\mathcal{O}(\alpha^{-2})$ using a similar argument as in (Chizat & Bach, 2018b, Lemma B2) for the present framework. We include the proof of this lemma here as the assumptions are not identical and to make the paper self-contained, while we do not claim any improvement on that result. To enounce this result, we recall that $\sigma_{\min}^{D\psi^{-1}}$ denotes the smallest singular eigenvalue of $D\psi^{-1}$ in a ball $\mathcal{B}_\delta^0(0)$, which is bounded away from 0 for $\delta$ small enough. Similarly, we recall that $\bar{g}_t^{-1} \succeq \sigma_{\min}^g \mathbb{1}$ for $\sigma_{\min}^g > 0$ in $\mathcal{B}_\delta^0(0)$ for $\delta$ small enough.

**Lemma A.2.** *Let $\overline{V}_t$, $\overline{\mathcal{V}}_t$ in $\overline{\mathcal{F}}_0$ be solutions of*

$$\frac{\mathrm{d}}{\mathrm{d}t}\overline{V}_t = \bar{g}_t^{-1}(D\psi_{\overline{V}_t}^{-1})^\top \Gamma(T^\lambda \alpha\psi^{-1}(\overline{V}_t) - \alpha\psi^{-1}(\overline{V}_t)),$$

$$\frac{\mathrm{d}}{\mathrm{d}t}\overline{\mathcal{V}}_t = \bar{g}_0^{-1}(D\psi_0^{-1})^\top \Gamma(T^\lambda \alpha\bar{\psi}^{-1}(\overline{\mathcal{V}}_t) - \alpha\bar{\psi}^{-1}(\overline{\mathcal{V}}_t)).$$

*Then defining $K := \sup_{t>0} \|(\bar{g}_t^{-1} - \bar{g}_0^{-1})(D\psi_{\overline{V}_t}^{-1})^\top \Gamma(T^\lambda \alpha\psi^{-1}(\overline{V}_t) - \alpha\psi^{-1}(\overline{V}_t))\|$ and $\beta := \frac{1-\gamma}{\kappa^2}(\sigma_{\min}^{D\psi^{-1}})^2$ we have that*

$$\sup_{t>0} \|\overline{V}_t - \overline{\mathcal{V}}_t\|_0 \le \frac{1}{\alpha}\frac{2K}{\beta}.$$

*Proof of Lemma A.2.* We define the function $h(t) := \frac{1}{2}\|\overline{V}_t - \overline{\mathcal{V}}_t\|_0^2$, take its time derivative

$$h'(t) = \langle \overline{V}_t' - \overline{\mathcal{V}}_t', \overline{V}_t - \overline{\mathcal{V}}_t \rangle_0,$$

and defining

$$(TD)_t := T^\lambda \alpha\psi^{-1}(\overline{V}_t) - \alpha\psi^{-1}(\overline{V}_t),$$

$$(\mathcal{TD})_t := T^\lambda \alpha \bar{\psi}^{-1}(\overline{\mathcal{V}}_t) - \alpha \bar{\psi}^{-1}(\overline{\mathcal{V}}_t)\,,$$

we evaluate (for simplicity of notation, we introduce the short hand $D\psi_t^{-1} := D\psi_{\overline{V}_t}^{-1}$ for the rest of the proof)

$$\overline{V}_t' - \overline{\mathcal{V}}_t' = \frac{1}{\alpha}\bar{g}_t^{-1}(D\psi_t^{-1})^\top \Gamma(TD)_t - \frac{1}{\alpha}\bar{g}_0^{-1}(D\psi_0^{-1})^\top \Gamma(\mathcal{TD})_t$$

$$\leq \frac{1}{\alpha}\left[\bar{g}_0^{-1}(D\psi_t^{-1})^\top \Gamma(TD)_t - \bar{g}_0^{-1}(D\psi_0^{-1})^\top \Gamma(\mathcal{TD})_t\right] \tag{A.39}$$

$$+ \frac{1}{\alpha}\left[\bar{g}_t^{-1}(D\psi_t^{-1})^\top \Gamma(TD)_t - \bar{g}_0^{-1}(D\psi_t^{-1})^\top \Gamma(TD)_t\right]\,. \tag{A.40}$$

We look at the two terms on the RHS separately and obtain, for (A.39)

$$\frac{1}{\alpha}\langle \bar{g}_0^{-1}(D\psi_t^{-1})^\top \Gamma(TD)_t - \bar{g}_0^{-1}(D\psi_0^{-1})^\top \Gamma(\mathcal{TD})_t, \overline{V}_t - \overline{\mathcal{V}}_t\rangle_0 \tag{A.41}$$

$$= \frac{1}{\alpha}\langle (D\psi_t^{-1})^\top \Gamma(TD)_t - (D\psi_0^{-1})^\top \Gamma(\mathcal{TD})_t, \overline{V}_t - \overline{\mathcal{V}}_t\rangle$$

$$= \frac{1}{\alpha}\langle (TD)_t - (\mathcal{TD})_t, D\psi_0^{-1}(\overline{V}_t - \overline{\mathcal{V}}_t)\rangle_\mu \tag{A.42}$$

$$+ \frac{1}{\alpha}\langle (D\psi_t^{-1} - D\psi_0^{-1})^\top \Gamma(TD)_t, \overline{V}_t - \overline{\mathcal{V}}_t\rangle\,. \tag{A.43}$$

We immediately see that by Lipschitz smoothness of $\psi^{-1}$ and the equivalence of $\|\cdot\|_2$ and $\|\cdot\|_0$ norms on $\Pi_r$ and (A.38), for (A.43) we have

$$\frac{1}{\alpha}\langle (D\psi_t^{-1} - D\psi_0^{-1})^\top \Gamma(TD)_t, \overline{V}_t - \overline{\mathcal{V}}_t\rangle \leq \frac{1}{\alpha}L_{D\psi^{-1}}\|\overline{V}_t\|_2\|\Gamma(TD)_t\|\|\overline{V}_t - \overline{\mathcal{V}}_t\|_2 \leq \frac{C_1}{\alpha^2}\sqrt{2h(t)}\,, \tag{A.44}$$

by choosing $C_1$ large enough. For (A.42) by the definition of $\psi$ we have

$$(TD)_t - (\mathcal{TD})_t = T^\lambda \alpha \psi^{-1}(\overline{V}_t) - T^\lambda \alpha \bar{\psi}^{-1}(\overline{\mathcal{V}}_t) - \alpha(\psi^{-1}(\overline{V}_t) - \bar{\psi}^{-1}(\overline{\mathcal{V}}_t))$$
$$= \alpha(P^\lambda - \mathbb{1})(\psi^{-1}(\overline{V}_t) - \bar{\psi}^{-1}(\overline{\mathcal{V}}_t))\,,$$

and hence, by (A.35) we have

$$\frac{1}{\alpha}\langle (TD)_t - (\mathcal{TD})_t, D\psi_0^{-1}(\overline{V}_t - \overline{\mathcal{V}}_t)\rangle_\mu \leq \langle (P^\lambda - \mathbb{1})(\psi^{-1}(\overline{V}_t) - \bar{\psi}^{-1}(\overline{\mathcal{V}}_t)), D\psi_0^{-1}(\overline{V}_t - \overline{\mathcal{V}}_t)\rangle_\mu$$

$$\leq \langle (P^\lambda - 1)D\psi_0^{-1}(\overline{V}_t - \overline{\mathcal{V}}_t), D\psi_0^{-1}(\overline{V}_t - \overline{\mathcal{V}}_t)\rangle_\mu$$
$$+ L_{D\psi^{-1}}\|\overline{V}_t\|_\mu^2\|D\psi_0^{-1}\|\|\overline{V}_t - \overline{\mathcal{V}}_t\|_\mu\,.$$

Defining $\beta := \frac{1-\gamma}{\kappa^2}(\sigma_{\min}^{D\psi^{-1}})^2$, the first term from above can be bounded as in (A.18) to obtain

$$\langle (P^\lambda - 1)D\psi_0^{-1}(\overline{V}_t - \overline{\mathcal{V}}_t), D\psi_0^{-1}(\overline{V}_t - \overline{\mathcal{V}}_t)\rangle_\mu \leq -\beta h(t)\,, \tag{A.45}$$

while for the second by our choice of $\delta = C/\alpha$ we have

$$L_{D\psi^{-1}}\|\overline{V}_t\|_\mu^2\|D\psi_0^{-1}\|\|\overline{V}_t - \overline{\mathcal{V}}_t\|_\mu \leq \frac{C^2}{\alpha^2}\kappa L_{D\psi^{-1}}\|D\psi_0^{-1}\|\sqrt{2h(t)}\,. \tag{A.46}$$

Finally, combining (A.44), (A.45) and (A.46) we have

$$(A.41) \leq -\beta h(t) + \frac{C_2}{\alpha^2}\sqrt{2h(t)}\,, \tag{A.47}$$

where $C_2 := C_1 + C^2\kappa L_{D\psi^{-1}}\|D\psi_0^{-1}\|$.
We now consider (A.40). Here by the definition of $K$ we have

$$\frac{1}{\alpha}\langle (\bar{g}_t^{-1} - \bar{g}_0^{-1})D\psi_t^{-1}\Gamma(TD)_t, \overline{V}_t - \overline{\mathcal{V}}_t\rangle_0 \leq \frac{K}{\alpha}\|\overline{V}_t - \overline{\mathcal{V}}_t\|_0 = \frac{K}{\alpha}\sqrt{2h(t)}\,.$$

Combining the above with (A.47) we finally obtain

$$h'(t) \leq -\beta h(t) + \frac{K}{\alpha}\sqrt{2h(t)} + \frac{C_2}{\alpha^2}\sqrt{2h(t)} \leq -\beta h(t) + \frac{2K}{\alpha}\sqrt{h(t)}\,,$$

for $\alpha$ large enough. The above expression is negative as soon as $h(t) > 4K^2/(\alpha\beta)^2$. Therefore, because $h(0) = 0$, we must have that $h(t) \leq 4K^2/(\alpha\beta)^2$ for all $t > 0$, *i.e.*,

$$\|\overline{V}_t - \overline{\mathcal{V}}_t\|_0 < \frac{1}{\alpha}\frac{2K}{\beta} \qquad \text{for all } t > 0,$$

as claimed. $\qquad\square$

To achieve the claimed $\mathcal{O}(\alpha^{-2})$ bound, we observe that $K$ in the above Lemma can be chosen $\mathcal{O}(\alpha^{-1})$ by the Lipschitz continuity of $\bar{g}_t^{-1}$. Indeed, since we chose $\|\overline{V}\|_0 = C/\alpha$, by (A.38) we have that

$$K \leq \sup_{t>0}\|\Gamma(TD)_t\|\|D\psi_{\overline{V}_t}^{-1}\|L_{\bar{g}_0^{-1}}\|\overline{V}\|_0 \leq C_0\sigma_{\max}^{D\psi^{-1}}L_{\bar{g}_0^{-1}}\frac{C}{\alpha} \leq \frac{\beta}{2}\frac{K'}{\alpha},$$

for $K'$ large enough, and therefore

$$\|\overline{V}_t - \overline{\mathcal{V}}_t\|_0 < \frac{K'}{\alpha^2} \qquad \text{for all } t > 0. \tag{A.48}$$

**Mapping to the embedding space.** We conclude the proof by mapping back to the original space, where we have

$$\begin{aligned}
\sup_{t>0}\|\mathcal{V}_t - V_t\|_\mu &= \sup_t\|\alpha\psi^{-1}(\overline{V}_t) - \alpha\bar{\psi}^{-1}(\overline{\mathcal{V}}_t)\|_\mu \\
&\leq \sup_t \alpha\left(\|D\psi_0^{-1}(\overline{V}_t - \overline{\mathcal{V}}_t)\|_\mu + L_{D\psi^{-1}}\|\overline{V}_t\|_\mu^2\right) \\
&\leq \alpha\left(\kappa\|D\psi_0^{-1}\|\sup_t\|\overline{V}_t - \overline{\mathcal{V}}_t\|_0 + \kappa^2 L_{D\psi^{-1}}\sup_t\|\overline{V}_t\|_0^2\right).
\end{aligned}$$

Then, letting $\mathcal{V}^*$ be the fixed point of (A.30) (unique and attracting by Tsitsiklis & Van Roy (1997)), by our choice of $\delta = C/\alpha$, (A.32) and (A.48) we have that

$$\begin{aligned}
\|\tilde{V}^* - V^*\|_\mu &\leq \|\mathcal{V}^* - V^*\|_\mu + \sup_{t>0}\|\mathcal{V}_t - V_t\|_\mu \\
&\leq \frac{1-\gamma\lambda}{1-\gamma}\|\Pi_0 V^* - V^*\|_\mu + \frac{1}{\alpha}(\kappa\|D\psi_0^{-1}\|K' + \kappa^2 L_{D\psi^{-1}}C^2),
\end{aligned}$$

as claimed. $\qquad\square$

