# OpenReview forum: "Temporal-difference learning for nonlinear value function approximation in the lazy training regime"
_ICLR.cc/2020/Conference — Reject_

### Official Review · AnonReviewer1 · 2019-10-15
**Official Blind Review #1**

**Rating:** 6

**Review:**

This paper discussed the nonlinear value function approximation for temporal different learning on on-policy policy evaluation tasks in the lazy training regime. The authors proved that for the both over-parameterized case (state number is fixed finite number) and the under-parameterized case (state number is sufficiently large or infinite), the value function can converge to some stationary point with exponential convergence rate. Moreover, the authors also characterized the error when the value function is under-parameterized.

Overall, this is a good paper. But the paper organization is awful. There are many places that are ambiguous or with notations that not formally defined. It may not due to the page limit as the authors currently use only 8 pages. I think the authors should polish the whole paper and make it more readable. Below are some main clarity issues I find, but the authors should not only solve the issues I mentioned.

1. For better presentation, I suggest the authors include a notation paragraph in the main text, which will be very helpful for the readers.
2. I think it would be better to mention (7) returns w that V_w = V^* / \alpha in Sec. 2 for better reading.
3. In Theorem 3.5, it is unclear that the estimation \tilde{V}^* is from \alpha V_{w}.
4. The WP1 in the paragraph after Theorem 3.5 means with probability 1?
5. In Equation (11), what is the definition of the measure \pi? If I understand correctly it is still \mu as the invariant measure should be fixed for a given policy?
6. The last paragraph in the proof of Theorem 3.5 is hard to follow. It can be better to introduce the result from the textbook and list the condition that need to verify, then give the lemmas show that the conditions can be verified.
What is the functional X in (12)? Should mention it in the main text, not in the appendix.
7. Figure 2 is somehow hard to understand. Maybe better show how the projection of TD error vector field outwards along the spiral in (a) and inwards in (b) in the figure.

From my point of view, the proof is almost correct and most of the lemmas are standard. This result is meaningful as it shows how and when the nonlinear function approximation will converge in temporal-difference learning (under the context of lazy training, and I think it is also correct under the context of NTK). The perspective on viewing the TD learning as linear dynamic system on functional space can inspire several new research in this field. My main concern is about the paper organization. I feel it can be hard for the potential readers to go through the whole paper. If the authors improve the quality of writing during the rebuttal period, I will consider improve my score.

**Experience Assessment:**

I have read many papers in this area.

**Review Assessment: Checking Correctness Of Derivations And Theory:**

I assessed the sensibility of the derivations and theory.

**Review Assessment: Checking Correctness Of Experiments:**

I did not assess the experiments.

**Review Assessment: Thoroughness In Paper Reading:**

I read the paper at least twice and used my best judgement in assessing the paper.

---

> ### Author Response · Authors · 2019-11-14
> **Response to referee #1**
>
>
> We thank the referee and we acknowledge that they have liked the content of our paper. We will gladly address the remarks of the referee about the form, and will introduce a notation section with the residual space we have at our disposal. We further clarify the following points:
>
> - $\pi_r$ is a map between $\bar{W_0}$ and $\bar{F_0}$ defined at the end of page 5 and does not coincide with the measure $\mu$
> - the WP1 does indeed mean "with probability one" and is now made clear,
>
> - the vector field $X$ is defined in the main text (and not only in the appendix) as the RHS of (11) in the third line of the third paragraph on page 6,
>
> - we appreciate the comment on the projection of the vector field, which we will implement in the final version of the paper, and will clarify the use of Lemmas 3.6 and 3.7 in the last paragraph of the proof of Thm 3.5.
>
> We thank the referee for suggestions on the organization of the paper and will follow them to improve the readability of the paper. We understand that some arguments may be hard to follow due to their technicality, and will do our best to improve on that while maintaining the mathematical rigor of the paper.

---

> > ### Comment · AnonReviewer1 · 2019-11-14
> > **Thanks for your response. Below are my ideas.**
> >
> > For $\pi$ in Equation (11) I mean the measure the authors used for the inner product $\langle \cdot, \cdot \rangle_{\pi}$, not the $\pi_r$. I don't find the definition of the corresponding $\pi$, please have a look.
> >
> > I agree with the authors that it is hard to balance the mathematical rigor and readability. From my point of view, I like the authors' writing style with rigorous mathematical description. However, we still need to consider the way to present the results, as I think most of the researchers in this field are not familiar with those mathematical concepts, e.g. operators, manifolds, geodesics etc. (though they should have some basic concept I think). So I suggest the authors continuously improve the writing. Maybe the authors can have a look at the style of [1].
> >
> > [1] Arora, S., Du, S., Hu, W., Li, Z. & Wang, R.. (2019). Fine-Grained Analysis of Optimization and Generalization for Overparameterized Two-Layer Neural Networks. Proceedings of the 36th International Conference on Machine Learning, in PMLR 97:322-332

---

> > > ### Author Response · Authors · 2019-11-15
> > > **second response to referee #1**
> > >
> > > We warmly thank the referee for a second insightful round of comments.
> > >
> > > Indeed, the scalar product in (11) is weighted by the invariant measure, so by $\mu$ instead of $\pi$. We have corrected the typo in the paper.
> > >
> > > We will also do our very best to continuously improve our style as suggested by the referee in order to make the this and future papers more accessible for a public that is less familiar with concepts of differential geometry, while respecting the page limit of our submission.

---

### Official Review · AnonReviewer2 · 2019-10-23
**Official Blind Review #2**

**Rating:** 3

**Review:**

The paper considers TD learning with function approximation, and establishes convergence results for over- and under-parameterized models in the lazy regime, and illustrates the theory on simple numerical examples.

Although the obtained results are interesting and the paper is well written, the contribution is quite incremental, in that it simply combines prior work on TD learning with linear function approximation with lazy training in order to show that models with a certain scaling can lead to convergence. This scaling makes the models essentially linear in the parameters (with a non-linear feature map given by initialization), so that it is not surprising that convergence can be reached, given the prior work on linear function approximation. I encourage the authors to further explain their motivation in studying such a setting.

It is claimed that the over-parameterized regime is only useful for finite state spaces, which seems quite limiting, since one cannot obtain global convergence in the under-parameterized case. When considering neural networks at infinite width, would the results be applicable if one assumes that V* belongs to the RKHS of the corresponding neural tangent kernel?

**Experience Assessment:**

I have read many papers in this area.

**Review Assessment: Checking Correctness Of Derivations And Theory:**

I did not assess the derivations or theory.

**Review Assessment: Checking Correctness Of Experiments:**

I assessed the sensibility of the experiments.

**Review Assessment: Thoroughness In Paper Reading:**

I read the paper at least twice and used my best judgement in assessing the paper.

---

> ### Author Response · Authors · 2019-11-14
> **Response to referee #2**
>
> We thank the referee for the review and appreciate their comment on a possible extension of our work, which we see as an interesting direction for future research. As we point out in the paper, we do see the the overparametrized regime as a simpler setting to establish stronger results while developing the intuition of the reader, in preparation for the more interesting but significantly more involved situation of the underparametrized case.
> Concerning the novelty of the paper, we believe that the deep reinforcement learning community would benefit from results ensuring convergence of popular algorithms such as deep neural networks with Xavier initialization in the large width regime in the context of value function approximation, as these were absent in the present literature. Furthermore, our results extend previous ones in the supervised setting to a different framework, including non-gradient dynamics (heavily leveraged in the previous results on nonlinear function approximation) and error estimates with the limiting kernel method and generalize those in the linear function approximation setting to include wide neural networks of any depth.

---

### Official Review · AnonReviewer3 · 2019-10-24
**Official Blind Review #3**

**Rating:** 3

**Review:**

This paper analyses the convergence of on-policy TD-learning for policy evaluation with non-linear function approximation (deep nets) in the lazy regime. Similar to deep learning theory, the key idea is that in the lazy regime, for an overparameterized network, if initialized in a certain manner, weights do not change significantly during training. The paper heavily draws upon techniques from n Chizat & Bach (2018) and adapts them to the setting with value functions, and policy evaluation. In order to get a strongly convex objective in function space, they consider a strongly convex Lyapunov function for the analysis. In the under-parameterized regime, the paper shows convergence to local optimum, by showing that convergence is exponentially fast to a local minimum where the key insight is that standard differential geometry can be applied to analyze the behavior of the projection on top of TD-lambda operator using past existing analyses. Finally, Section 4 shows numerical examples -- a divergent function approximator and some empirical results on a single-layer neural net.

Overall, I lean in favor of rejection for this paper. I am mainly concerned with the significance of the content in the paper and positioning with respect to past theoretical/empirical work. My specific concerns are:

1. Assumption 1 is strong, it assumes full support over the state-space, however, in any situation of practical relevance, this is not the case. There are many papers at this point (for example, [1], [2]) which show empirically that even with Q-learning (Fitted Q), divergence is not common if the function approximator is large/wide enough, and the support of the state space is full, however, these methods can diverge if the support is not full/ skewed. I am not sure if this assumption is very realistic then.  (The results are for specific function approximation in this paper, and hence it is unclear if that is the case we use in practice)

2. I am not sure if the techniques used in the paper are relatively novel (from a theoretical point of view), and I would appreciate if the authors can elaborate a bit on this. It seems like most of the proof is drawn from past work (although past work has theoretical results in a different problem setting -- supervised learning). While the setting of policy evaluation is novel, I am concerned about how many new techniques are to be gained from a theoretical point of view here.

3. What assumptions are needed for Theorem 3.1 and Theorem 3.2, from a deep-learning standpoint? What recommendations does the theory give to practitioners? I find a discussion on both of these points missing from the paper. It would be appreciated if the authors can elaborate on these points.

References:
[1] Deep Reinforcement Learning and the Deadly Triad, van Hasselt et.al.
[2] Diagnosing Bottlenecks in Deep Q-learning Algorithms, Fu et.al.

**Experience Assessment:**

I have read many papers in this area.

**Review Assessment: Checking Correctness Of Derivations And Theory:**

I assessed the sensibility of the derivations and theory.

**Review Assessment: Checking Correctness Of Experiments:**

I assessed the sensibility of the experiments.

**Review Assessment: Thoroughness In Paper Reading:**

I read the paper at least twice and used my best judgement in assessing the paper.

---

> ### Author Response · Authors · 2019-11-14
> **Response to referee #3**
>
> We thank the refereee for the comments, which we address below.
>
> 1) Assumption 1 ensures that the underlying Markov Reward Process (MRP) is ergodic, and as the reviewer points out it is a common assumption in the literature. The reason for this assumption is that it guarantees sufficient exploration of the phase speace, a fundamental requirement for the convergence of reinforcement learning algorithms. Indeed, if the MRP does not visit all the states with positive probability, then it may never visit some key states for the evaluation of the value function, and the algorithm would therefore necessarily converge to a wrong estimate of such function. We therefore don't think it is reasonable to relax this assumption in our setting.
>
> 2) The results obtained in our paper combine results and techniques from various previous works, improving them in various aspects, for example extending the techniques developed in [COB19] to a non-gradient setting (a structure heavily leveraged in that paper) and quantifying the error between the nonlinear approximator and the limiting, linear one. As guarantees of convergence are of fundamental importance in the setting of deep reinforcement learning, and since the rationale developed in this paper has a wide applicability, in particular in that subject, we believe that the results presented here are relevant in the setting of RL.
>
> 3) The theory ensures for example that the population dynamics of Temporal-Difference learning using deep neural networks of a sufficient width (and any depth), initialized in the Xavier regime is convergent on with probability one to a global minimum in the discrete setting, and not divergent in the continuous setting. We believe this to be an example of interest for practicionners and a novel result.
>
> [COB19] Chizat, Oyallon, Bach ``On Lazy Training in Differentiable Programming'', 2019

---

### Official Review · AnonReviewer4 · 2019-11-27
**Official Blind Review #4**

**Rating:** 6

**Review:**

The paper discusses the policy evaluation problem using temporal-difference (TD) learning with nonlinear function approximation. The authors show that in the “lazy training” regime both over- and under-parametrized approximators converge exponentially fast, the former to the global minimum of the projected TD error and the latter to a local minimizer of the same error surface. Simply put, the lazy regime refers to the approximator behaving as if it had been linearized around its initialization point. This can happen if the approximation is rescaled, but can also occur as a side-effect of its initialization. The authors present simple numerical examples illustrating their claims.

Although I did not carefully check the math, this seems like a solid contribution on the technical side. My main concern about the paper is that it falls short in providing intuition and contextualizing its technical content. Regarding the presentation, I believe it is possible to have a less dry prose without sacrificing mathematical rigor. If some of the technical material is moved to the appendix --like auxiliary results, discussion on proof techniques, etc--, the additional space could be used to discuss the implications of the theoretical results in more accessible terms.

For example, a subject that ought to be discussed more clearly is the nature of the approximation induced by the lazy training regime. As far as I understand, this regime can be thought of as a sort of regularization that severely limits the capacity of the approximator.  Although the authors mention in the conclusion that “...convergence of lazy models may come at the expense of their expressivity”, after reading the paper I do not have a clear sense of how expressive such models actually are. In their experiments, Chizat et al. (2018) observed that the performance of commonly used neural networks degrades when trained in the lazy regime --to a point that they consider it unlikely that the successes of deep learning can be credited to this regime of training. It seems to me that this subject should be more explicitly discussed in a paper that sets out to provide theoretical support for deep reinforcement learning.

Still regarding the behavior of lazy approximators, my intuitive understanding is that they work as a linear model using random features. If this interpretation is correct, this makes the theoretical results a bit less surprising. They are still interesting, though, for they can be seen as relying on a “smoother” version of the linearity assumption often made in the related literature. Maybe this is also something worth discussing? Still on this subject, it seems to me that one potential disadvantage of lazy models with respect to their linear counterparts is that it is less clear how to enforce the lazy regime in practice. In Section 4.2 the authors discuss how this can naturally happen as a side-effect of the initialization, but it is unclear how applicable the particular strategy used to illustrate this phenomenon, with the “doubling trick”, is in practice. This is another example of a less technical discussion that would make the paper a stronger contribution.

**Experience Assessment:**

I have read many papers in this area.

**Review Assessment: Checking Correctness Of Derivations And Theory:**

I did not assess the derivations or theory.

**Review Assessment: Checking Correctness Of Experiments:**

I did not assess the experiments.

**Review Assessment: Thoroughness In Paper Reading:**

I read the paper at least twice and used my best judgement in assessing the paper.

---

### Decision · Program_Chairs · 2019-12-19

**Decision:**

Reject

**Comment:**

This paper provides convergence results for Non-linear TD under lazy training.

This paper tackles the important and challenging task of improving our theoretical understanding of deep RL. We have lots of empirical evidence Q-learning and TD can work with NNs, and even empirical work that attempts to characterize when we should expect it to fail. Such empirical work is always limited and we need theory to supplement our empirical knowledge. This paper attempts to extend recent theoretical work on the convergence of supervised training of NN to the policy evaluation setting with TD.

The main issue revolves around the presentation of the work. The reviewers found the paper difficult to read (ok for theory work). But, the paper did not clearly discuss and characterize the significance of the work: how limited is the lazy training regime, when would it be useful? Now that we have this result, do we have any more insights for algorithm design (improving nonlinear TD), or comments about when we expect NN policy evaluation to work?

This all reads like: the paper needs a better intro and discussion of the implications and limitations of the results, and indeed this is what the reviewers were looking for. Unfortunately the author response and paper submitted were lacking in this respect. Even the strongest advocates of the work found it severely lacking explanation and discussion.  They felt that the paper could be accepted, but only after extensive revision.

The direction of the work is important. The work is novel, and not a small undertaking. However, to be published the authors should spend more time explaining the framework, the results, and the limitations to the reader.